# EA-PS: Estimated Attack Effectiveness based Poisoning Defense in Federated Learning under Parameter Constraint Strategy

## Abstract

Federated learning is vulnerable to poisoning attacks due to the characteristics of its learning paradigm. There are a number of server-based and client-based backdoor defense methods to mitigate the impact of the attack. However, when facing persistent adaptive attacks with long-lasting attack effects, defense methods fail to guarantee robust and stable performance. In this paper, we propose a client-side defense method, EA-PS, which can be effectively combined with server-side methods to address the above issues. The key idea of EA-PS is to constrain the perturbation range of local parameters while minimizing the impact of attacks. To theoretically guarantee the performance and robustness of EA-PS, we prove that our methods have an efficiency guarantee with a lower upper bound, a robustness guarantee with a smaller certified radius, and a larger convergence upper bound. Experimental results show that, compared with other client-side defense methods combined with different server-side defense methods under both IID and non-IID data distributions, EA-PS achieves lower attack success rates and more stable defense performance with smaller variance. Our code can be found at **https://anonymous.4open.science/ r/EA-SP-6BC9**.

## 1. Introduction

Federated learning (FL) (Huang et al., 2023b) is a distributed machine learning paradigm that enables multiple parties to train models while preserving data privacy and security collaboratively. However, due to its decentralized nature, FL is vulnerable to attacks, particularly when clients are

[1] Anonymous Institution, Anonymous City, Anonymous Region, Anonymous Country. Correspondence to: Anonymous Author <anon.email@domain.com>.

Preliminary work. Under review by the International Conference on Machine Learning (ICML). Do not distribute.

compromised. Numerous studies (Wu et al., 2022; Lyu et al., 2023; Li et al., 2022) have shown that malicious clients can manipulate the global model, a type of attack usually known as poisoning attacks. Such attacks (Lyu et al., 2020) have the potential to degrade the accuracy level of the model, lead to incorrect predictions, and result in significant damage.

Various defense strategies have been proposed to mitigate the impact of these attacks on the server side, such as CMA&CTMA (Yin et al., 2018), Multi-Krum (Mhamdi et al., 2018a) and Bulyan (Blanchard et al., 2017). However, these server-side defense methods fail to withstand strong attacks (Zhu et al., 2023), such as adaptive attacks (Sun et al., 2019) and persistent backdoor attacks (Liu et al., 2024) with long-lasting attack effects (Sun et al., 2021). To tackle the aforementioned issue, client-side defense methods provide more effective protection performance, combined with server-side defense methods. FL-WBC (Sun et al., 2021) employs perturbations for defense, but the randomness of these perturbations can lead to a worse backdoor defense rate. To minimize the effect of attacks, LeadFL (Zhu et al., 2023) enhances FL-WBC (Sun et al., 2021) by utilizing hessian matrix optimization techniques. In this paper, we introduce an enhanced objective function that showcases superior defense performance with a smaller upper bound when compared with LeadFL.

We empirically show that EA-PS$^-$ (LeadFL with our objective function) has lower backdoor accuracy than FL-WBC (Sun et al., 2021) and LeadFL with various server-side defense methods in IID and non-IID settings, as shown in Figure 1. More importantly, we observe that the backdoor defense performance of all three methods is unstable with large backdoor accuracy variances and distribution intervals. Therefore, an additional defense method is needed to enhance the stability of backdoor defense with lower backdoor accuracy, which is another goal of this paper.

Therefore, we propose a client-based defense approach named Estimated Attack Effectiveness based Poisoning Defense method under Parameter Constraint Strategy (EA-PS). It minimizes the long-lasting backdoor attack effect with a parameter constraint strategy to enhance stability by constraining the perturb range in the parameter space. We derive that our method has a smaller optimization

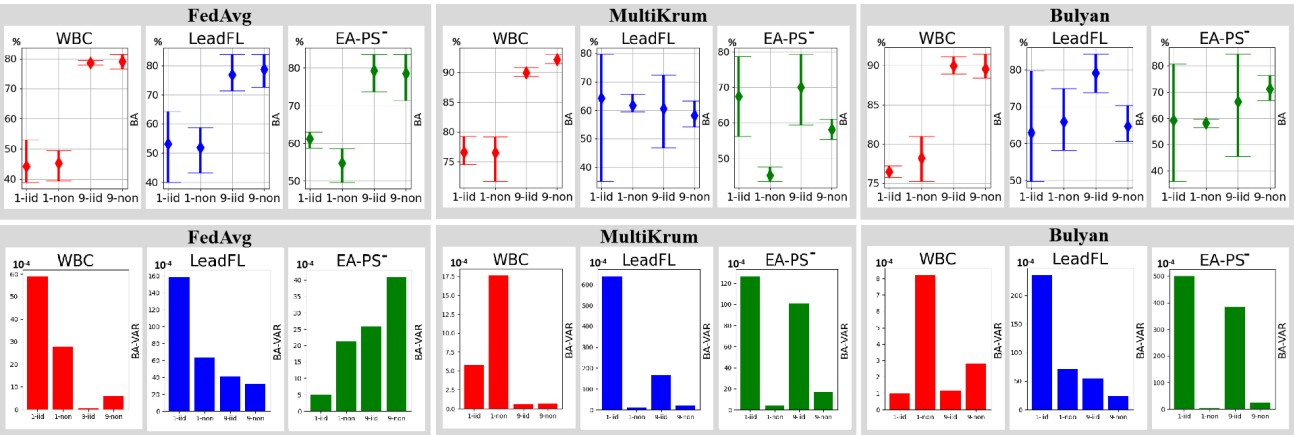

*Figure 1.* Defense performance of server-side defense methods under different attacks and data distributions on CIFAR10 dataset. The upper images are about the performance intervals of BA, and the other images are about the performance variances of BA.

upper bound and certified radius. Then, through Lagrangian relaxation and linear robust optimization, we integrate the constraints into the loss function to obtain an approximately optimal solution. Finally, by using a regularization method on parameter constraint, we increase the convergence upper bound while adaptively limiting the disturbance range of the parameter space. We evaluate our defense methods on Fashion-MNIST and CIFAR10 against the model poisoning attack under IID and non-IID settings. The results demonstrate that `EA-PS` can effectively mitigate the attack effect with stable defense performance.

Our key contributions are summarized as follows:

- We designed the `EA-PS` method, which effectively defends against poisoning attacks by minimizing the impact of long-lasting attacks and ensures the stability of the backdoor defense effect by the parameter constraint strategy.

- We derive a lower theoretical upper bound of the enhanced objective function to prove the efficiency of the `EA-PS` method. Moreover, when implementing the `EA-PS` method, we also derive a robustness guarantee featuring a smaller certified radius and a larger convergence upper bound.

- We evaluate our defense methods on FashionMNIST and CIFAR10 datasets under IID and non-IID settings against the model poisoning attacks with different server-side defense methods. The results show that our proposed defense methods can enhance the robustness of FL with a lower attack success rate by up to 14.9% and more stable defense performance with smaller variance by up to 40% compared with other client-side defense methods.

## 2. Related Work

### 2.1. Poisoning Attack in FL

Model poisoning attacks can be classified into untargeted attacks (Li et al., 2022; Lian et al., 2023b) and targeted attacks (Wu et al., 2022; Lyu et al., 2023). The objective of untargeted attacks is to disrupt the prediction accuracy of the model for any test input, while targeted attacks (also known as backdoor attacks) aim to misclassify samples with specific triggers into categories chosen by the attacker. Persistent attack strategies (Liu et al., 2024) and adaptive attack strategies (Zhang et al., 2023) are widely used to increase the success rate of backdoor attacks. Our approach specifically addresses targeted poisoning attacks (1/9-pixel attack) (Bagdasaryan & Shmatikov, 2020) with adaptive and persistent attack strategies (Liu et al., 2024; Zhang et al., 2023).

### 2.2. Prior Art on Defense Methods

Server-side defense methods for federated learning are generally classified into two main categories: outlier detection/filtering approaches (Huang et al., 2023a) and robust aggregation methods (Mhamdi et al., 2018b). The fundamental principle of filtering methods is to mitigate backdoor attacks by identifying and excluding malicious or anomalous client-side model updates. However, they may not be able to fully utilize the information from all clients, affecting model's performance (Li et al., 2019a). Robust aggregation techniques are designed to mitigate the effect of adversarial model updates on the global model by employing robust aggregation strategies to identify and discard malicious updates, ensuring that the global model's training process remains unaffected by the actions of compromised clients.

Client-side defense methods provide more powerful protec-

tion performance combined with server-side defense methods. Existing client defense methods are divided mainly into differential-privacy based methods (Naseri et al., 2020; Guo et al., 2024) and parameterized methods (Sun et al., 2021; Zhu et al., 2023). The effect of differential-privacy based methods is uncontrollable due to the uncertainty of the amount of noise (Lian et al., 2023a). While the parameterized approach suffers from the inability to achieve tighter upper bounds and instability in defense performance.

Therefore, we propose the EA-PS method, which achieves lower upper optimization bounds with smaller certified radius, and offers superior convergence properties and more stable defense performance compared to other methods.

## 3. Motivation

Although current server-side and client-side defense methods can protect models against poisoning attacks (Sun et al., 2021; Zhu et al., 2023), they struggle to maintain stable and robust performance under extremely strong persistent attacks with long-lasting attack effects. To investigate the performance of current state-of-the-art methods with persistent attacks, we measured backdoor accuracy (BA) and their variance (VAR) of different server-side defense methods (LeadFL (Zhu et al., 2023) and WBC (Sun et al., 2021)) combined with client-side defense methods (FedAvg (McMahan et al., 2016), MultiKrum (Blanchard et al., 2017) and Bulyan (Mhamdi et al., 2018a)) under different long-lasting attack settings (1/9 pixel attacks) on CIFAR10 dataset. The experimental results in Figure 1 are already discussed in Section 1. Details about the results can be found in Appendix C.

As shown in Figure 1, we can get two observations. 1) WBC (Sun et al., 2021) method, designed with gradient constraint, has the most stable performance with the worst BA performance; 2) LeadFL, designed with the constraint of gradient variation trend, has a better but not stable BA performance than WBC. Motivated by these, we design a new optimization method (Denoted as EA-PS⁻) with more historical information on the constraint of gradient variation trends, to minimize the impact of long-lasting attacks. The experimental results are shown in Figure 1 to verify that EA-PS⁻ is significantly improved compared with LeadFL, while still unstable.

In order to ensure the stability of the defense effect, the optimization space of the model needs to be constrained to maintain the stability of the model parameters, so we designed the parameter constraint strategy to ensure the stability of the defense performance. As shown in Figure 2, the simple idea of the parameter constraint strategy is to map the optimized manifold space of $A$ into the unit space $I$ by converting the spatial constraints into the base (rank) constraint $\lambda$ with spatial mapping $B$, reducing the

dimensionality of the constraint space, and improving the efficiency of the constraint by simplifying the complexity of the parameter constraints.

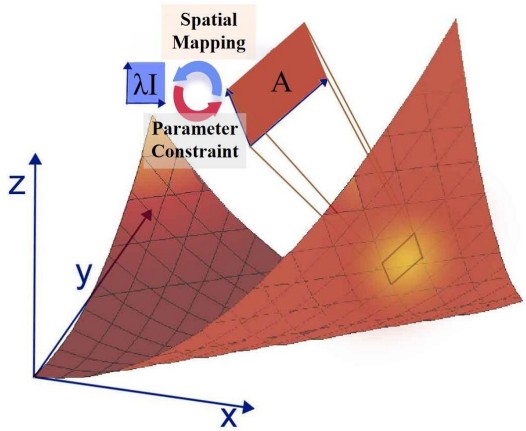

*Figure 2.* Illustration of the parameter constraint strategy.

## 4. Model Poisoning Attack in FL

To better understand the impact of model poisoning attacks in FL, we first analyze the impact of poisoning attacks and the relationship between attacks in different rounds (Sun et al., 2021). During this process, we have developed an optimized objective function with a lower upper bound than LeadFL (Zhu et al., 2023) and provided a detailed proof. However, we observe that the parameters are not stable. So, we provide the parameter constraint strategy to ensure parameter stability. Without losing generality, this paper adopts the most widely used FL algorithm, FedAvg (McMahan et al., 2016).

### 4.1. Problem Formulation

The aggregation objective of FedAvg is defined as follows:

$$\theta = \min_{\theta} \left\{ F(\theta) \triangleq \sum_{k=1}^{N} p^k F^k(\theta) \right\}, \qquad (1)$$

where $\theta$ is the weights of the global model, $N$ represents the number of devices, $F^k$ is the local objective of the $k$-th device, $p^k$ represents the weight of the $k$-th device.
In the $t$-th round of communication, the client updates the weights in the $e$-th round of local training as follows:

$$\theta_{t,e+1}^k \leftarrow \theta_{t,e}^k - \eta_{t,e} \triangledown F(\theta_{t,e}^k), \qquad (2)$$

where $\eta_{t,e}$ represents the learning rate and each local training round is updated on a mini-batch of data samples chosen from $k$-th client's data set. Finally, the server averages the parameters submitted by the $k$ models selected for aggrega-

tion (Zhu et al., 2023) as follows:

$$\theta^t \leftarrow \frac{N}{K} \sum_{k \in S_t} p^k \theta_t^k, \tag{3}$$

where $S_t$ is a set of participating clients in the $t$-th round. $K$ is the number of selected clients by server-side defense methods.

### 4.2. Long-lasting Attack Effect

Based on FL-WBC (Sun et al., 2021), define $\delta_t$ as the effect of the attack on the client in the $t$-th round as follows:

$$\delta_t = \frac{N}{K} \left[ \sum_{k \in \mathbb{S}_t} p^k \prod_{e=0}^{E-1} \left( \boldsymbol{I} - \eta_{t,e} \boldsymbol{H}_{t,e}^k \right) \right] \delta_{t-1}, \tag{4}$$

where $\boldsymbol{H}_{t,e}^k \triangleq \bigtriangledown^2 F(\theta_{t,e}^k)$ is the Hessian matrix at local iteration $e$ of global round $t$ and $I$ is the identify matrix.

For convenience, We define coefficient of attack impact $A_t$ as the relationship between two rounds as follows:

$$A_t \triangleq \sum_{k \in \mathbb{S}_t} p^k \prod_{e=0}^{E-1} \left( \boldsymbol{I} - \eta_{t,e} \boldsymbol{H}_{t,e}^k \right). \tag{5}$$

It follows from LeadFL (Zhu et al., 2023) that $A_t$ can be equated to:

$$A_t \sim I - (P_t - P_{t-1}) + \Delta_t, \tag{6}$$

where $P_t$ is the parameter for round $t$, $\Delta_t$ is updated for round $t$.

**Theorem 4.1.** *Minimizing $A_t - A_{t-1}$ yields a lower optimization upper bound than minimizing $A_{\hat{t}}$, where $A_{\hat{t}}$ is the coefficient of attack impact in LeadFL.*

**Proof 4.1.** From the definition of $A_t$, we can get Equations (7) as follows:

$$A_t = I - (P_t - P_{t-1}) + \Delta_t. \tag{7}$$

According to the Equation (7), we can obtain

$$A_t - A_{t-1} = 2P_{t-1} - P_t. \tag{8}$$

According to the Equation(8), we can obtain

$$P_t = 2P_{t-1} + \varepsilon, \tag{9}$$

$$P_t = \sum_{i=1}^{t} \Delta_i + P_0 + [(t-1) + \varepsilon_{(P_{t-1})}]\varepsilon, \tag{10}$$

where $\varepsilon_{(P_{t-1})}$ is the coefficients of $\varepsilon$ in the polynomial $P_{t-1}$.

By combining Equations (8) and (10), we can obtain

$$A_t = I - t\varepsilon. \tag{11}$$

From the definition of $A_{\hat{t}}$, we can get Equation (12) as follows:

$$A_{\hat{t}} = I - (P_{\hat{t}} - P_{\hat{t}-1}) + \Delta_{\hat{t}}. \tag{12}$$

According to the Equation (12), we can obtain

$$P_{\hat{t}} = P_0 + \hat{t}I + \sum_{k=1}^{\hat{t}} \Delta_k + \hat{t}\varepsilon. \tag{13}$$

By combining Equations (12) and (13), we can obtain

$$A_{\hat{t}} = I + \varepsilon. \tag{14}$$

By combining Equations (11) and (14), we can obtain

$$A_{\hat{t}} - A_t = (t+1)\varepsilon, \tag{15}$$

where $\varepsilon$ is the lower noise boundary. According to Equation (15), $A_t \leq A_{\hat{t}}$. The specific proof see Appendix A.2.

### 4.3. Parameter Constraint Strategy

From the observations in Figure 1, it can be noticed that only minimizing the coefficient of attack impact ($A_t - A_{t-1}$ and $A_t$ ) can lead to unstable backdoor defense performance. Therefore, we propose a parameter constraint strategy that constraints $A$ to a parameter boundary (denoted as $\lambda$) to ensure that certain specific attacks are effectively detected while ensuring the stability of the parameters, as described in Figure 2. The constraint equation is as follows:

$$\lambda I = B^{-1}AB, \tag{16}$$

$$AB = \lambda B, \tag{17}$$

where $B$ is equivalent to $\delta$ as the spatial mapping for the effects of the attack.

## 5. EA-PS

Within this section, we describe EA-PS, a robust client-side defense approach that can be arbitrarily combined with existing server-side defense approaches for a better and more stable defense performance.

### 5.1. Defense Design

The core idea of EA-PS is to better and more consistently eliminate the effects of attacks by minimizing the impact of attacks with stability. In particular, we reform the local model training of benign devices to achieve two goals:

- Goal 1: Minimize the impact of attacks to a better defense performance.

- Goal 2: Ensure the stability of defense performance by the parameter constraint strategy.

To achieve the first goal, we designed a new optimization on the constraint of gradient variation trends, to minimize the impact of long-lasting attacks, namely $A_t - A_{t-1}$. To achieve the second goal, we designed the parameter constraint strategy to ensure the stability of the defense effect, namely $AB = \lambda B$.

$$
\begin{aligned}
Obj. \quad & A_t - A_{t-1} \\
s.t. \quad & AB = \lambda B \\
& t > 1.
\end{aligned}
\tag{18}
$$

Since $AB = \lambda B$ is the constraint, we assume that parameter boundary $\lambda$ is a linear set of $A$ based on the linear decision rule (Bertsimas et al., 2019). Without loss of generality, we define the following set:

$$
\mathcal{L}^{T,N} = \left\{ \boldsymbol{A} \in \mathcal{R}^{T,N} \; \middle| \; \begin{array}{c} \exists \boldsymbol{A}_t, \boldsymbol{A}_{t-1}, \boldsymbol{t} \in [T_1] : \\ \lambda = \wp \boldsymbol{A}_t + \xi \boldsymbol{A}_{t-1} \end{array} \right\}, \tag{19}
$$

where $\wp$ and $\xi$ are auxiliary variables. Then, the problem becomes equation (20) with an upper bound approximation to the near-optimal solution of the model (Ben-Tal et al., 2004).

$$
\begin{aligned}
Obj. \quad & (A_t - A_{t-1}) + \alpha(AB - \lambda B) \\
s.t. \quad & \lambda \in \mathcal{L}^{T,N}, \\
& t > 1.
\end{aligned}
\tag{20}
$$

But, for convenience, we simplify and approximate the calculation by reducing equation (19) to $\lambda \simeq (A_t + A_{t-1})/2$. Then, we let $B$ adaptively change with coefficient $\beta$ to map better spatial space. We further denote $Regu$ as a regulation function to control the influence degree of $\beta$.

$$
\left\{
\begin{aligned}
& B = \beta B, \\
& Regu(\beta, B) = Max(\beta - \beta_{old}, 0.00001\beta),
\end{aligned}
\right.
$$

where $\beta_{old}$ is $\beta$ of the previous round.

Then, the problem is approximated to Equation (21) as follows,

$$
\begin{aligned}
Obj. \quad & (A_t - A_{t-1}) + \alpha(AB - \lambda B) + \gamma Regu(\beta, B) \\
s.t. \quad & \lambda \simeq (A_t + A_{t-1})/2 \\
& t > 1.
\end{aligned}
\tag{21}
$$

To ensure that the model can converge after the above process, gradient trimming is performed during local training with a threshold $q$.

$$
\operatorname{clip}\left( \nabla \left( \mathbf{I} - \eta_{t,e} \widetilde{\mathbf{H}}_{t,e}^k \right), q \right)_{r,c} =
$$
$$
\left\{
\begin{aligned}
& \nabla \left( \mathbf{I} - \eta_{t,e} \widetilde{\mathbf{H}}_{t,e}^k \right)_{r,c} \left| \nabla \left( \mathbf{I} - \eta_{t,e} \widetilde{\mathbf{H}}_{t,e}^k \right)_{r,c} \right| \le q, \\
& q, \left| \nabla \left( \mathbf{I} - \eta_{t,e} \widetilde{\mathbf{H}}_{t,e}^k \right)_{r,c} \right| > q,
\end{aligned}
\right.
$$

where $r$ and $c$ are the indexes of rows and columns.

---

**Algorithm 1** EA-PS and robust aggregation
___

**Input:** number of global rounds $T$, constraint rate $\alpha$, clipping bound $q$, ♯ of clients selected in a round $K$, dynamic coefficient of the spatial mapping $\beta$, regulation rate $\gamma$.

**for** communication round $t = 0$ **to** $T - 1$ **do**
  Server randomly chooses $K$ clients;
  **parallel** $k = 0 \ldots K$ **do**
  Update model weights as global weights from the last round;
  **for** local iteration $e = 0, 1, \ldots$ **do**
    Compute gradients and update weights:
    $\theta_{t,e+1}^k \leftarrow \theta_{t,e}^k - \eta_{t,e} \nabla F(\theta_{t,e}^k)$;
    Compute the effect of poisoning attack:
    $A_t = \sum_{k \in \mathbb{S}_t} p^k \prod_{e=0}^{E-1} \left( \boldsymbol{I} - \eta_{t,e} \boldsymbol{H}_{t,e}^k \right)$;
    Compute the parameter boundary:
    $\lambda = (A_t + A_{t-1})/2$;
    Minimize the effect of poisoning attack and constraint the boundary of parameter:
    $A_t - A_{t-1} + \alpha(AB - \lambda B) + \gamma Regu(\beta, B)$;
    Compute and clip gradients:
    $\boldsymbol{R}_{t,e}^k = \operatorname{clip}\left( \nabla \left( \boldsymbol{I} - \eta_t \widetilde{\boldsymbol{H}}_{t,e}^k \right), q \right)$;
    Update weights;
    Update dynamic coefficient of the spatial mapping:
    $\beta_{old} \leftarrow \beta$;
  **end for**
  Compute updates;
  **end parallel**
  Aggregate updates using server-side defense;
  Update weights;
**end for**

---

### 5.2. Convergence Analysis

In this subsection, we derive convergence guarantees for FedAvg using EA-PS in the context of no malicious model attack. Specifically, for the $t$-th round, the local model on the $k$-th benign device is updated as:

$$
\nabla F^{'}(\theta_{t,e}^k) = \nabla F(\theta_{t,e}^k) + \gamma Regu + clip. \tag{22}
$$

Based on Assumptions A.1 to A.5 of the Appendix, we can derive the convergence guarantee of our defense on FedAvg as follows.

**Theorem 5.1.** *(Convergence Guarantee): Let Assumptions A.1 to A.5 hold and $l, \mu, \sigma_k, G, K, N, \Gamma, F^*$ be defined therein and in Definition A.6. Choose $\kappa = \frac{l}{\mu}$, $\varphi = max(8\kappa, I)$ and the learning rate $\eta_t = \frac{2}{\mu(\varphi + t)}$. Then we have the following bound for* EA-PS:

$$
E[F(\theta_T)] - F^* \le \frac{\kappa}{\varphi + T - 1} \left( \frac{2(M+C)}{\mu} + \frac{\mu\varphi}{2} E[||\theta_0 - \theta^*||^2] \right), \tag{23}
$$

*Table 1.* Comparison of benign accuracy on FashionMNIST and CIFAR10 with IID/non-IID settings under 1/9-pixel backdoor attack.

| mean(%)/var($10^{-4}$) | | | FashionMNIST | | | | CIFAR10 | | | |
|---|---|---|---|---|---|---|---|---|---|---|
| | | | None | CMA | Multikrum | Bulyan | None | CMA | Multikrum | Bulyan |
| 9-pixel | IID | LeadFL | 93.22/7.75 | 95.2/2.8 | 32.82/23.4 | 32.78/99.34 | 76.9/41.04 | 78.34/14.66 | 60.63/166.51 | 79.16/54.75 |
| | | LeadFL$^+$ | 89.63/3.66 | 93.26/2.71 | 23.34/19.7 | 22.57/84.13 | 70.97/33.85 | 73.61/8.62 | 63.39/5.2 | 78.06/32.56 |
| | | EA-PS$^-$ | 88.8/1.86 | 92.73/6.91 | 21.73/52.12 | 19.7/246.97 | 79.35/25.91 | 62.05/35.68 | 32.12/100.78 | 66.37/383.42 |
| | | EA-PS | 90.72/3.02 | 91.93/4.23 | 22.97/7.95 | 19.59/65.19 | 77.8/6.69 | 63.57/0.01 | 62.75/0.38 | 61.3/0.04 |
| | non-IID | LeadFL | 92.43/2.29 | 91.93/8.37 | 17.38/28.33 | 15.14/61.37 | 78.78/32.38 | 61.21/11.06 | 58.13/21.63 | 64.65/25.71 |
| | | LeadFL$^+$ | 89.04/2.24 | 89.91/11.99 | 16.43/20.65 | 13.77/59.88 | 78/10.52 | 59.51/33.69 | 61.55/14.22 | 61.76/18.55 |
| | | EA-PS$^-$ | 88.1/13.98 | 89.9/18.91 | 12.58/17.4 | 11.38/93.83 | 78.57/40.99 | 63.66/18.54 | 53.42/16.76 | 71.11/23.79 |
| | | EA-PS | 86.67/2.15 | 91.82/0.68 | 11.79/4.76 | 12.45/22.72 | 77.1/5.52 | 63.21/3.35 | 52.2/8.78 | 66.05/15.58 |
| 1-pixel | IID | LeadFL | 89.83/2.49 | 89.53/2.76 | 60.5/64.4 | 58.72/215.5 | 53.09/158.59 | 46.54/301.31 | 64.26/640.64 | 63/236.4 |
| | | LeadFL$^+$ | 88.42/0.68 | 86.24/2 | 56/93.83 | 51.39/41.29 | 54.79/141.63 | 34.19/48.78 | 61.72/62.41 | 66.14/120.55 |
| | | EA-PS$^-$ | 86.24/20.7 | 85.48/6.52 | 49.87/42.56 | 45.45/237.83 | 61.14/4.9 | 44.71/427.22 | 67.4/126.97 | 59.15/500.37 |
| | | EA-PS | 88.76/4.67 | 84.84/3.34 | 45.32/17.94 | 45.68/189.3 | 50.6/15.6 | 32.58/0.02 | 52.5/15 | 59.11/7.61 |
| | non-IID | LeadFL | 88.52/0.54 | 91.27/5.03 | 47.5/66.29 | 32.32/46.74 | 51.84/63.03 | 31.47/2.35 | 61.74/11.71 | 65.96/71.8 |
| | | LeadFL$^+$ | 86.47/0.55 | 87.89/6.87 | 38.95/13.4 | 37.11/5.33 | 58.33/10.55 | 36.46/30.19 | 56.84/26.28 | 56.02/92.72 |
| | | EA-PS$^-$ | 86.06/7 | 85.52/4.42 | 34.97/101.25 | 27.29/29.88 | 54.74/21.37 | 37.66/83.36 | 45.2/4.54 | 58.19/2.63 |
| | | EA-PS | 86.46/6.06 | 84.96/0.26 | 35.27/ 32.24 | 22.13/18.3 | 54.52/34.95 | 33.18/91.01 | 44.68/2.31 | 60.74/7.22 |

*Table 2.* Comparison of different $\beta$ on CIFAR10 with IID/non-IID settings under 1/9-pixel backdoor attack.

| mean(%)/var($10^{-4}$) | | IID | | | | | non-IID | | | | |
|---|---|---|---|---|---|---|---|---|---|---|---|
| $\beta$ | | 0.01 | 0.03 | 0.05 | 0.07 | 0.09 | 0.01 | 0.03 | 0.05 | 0.07 | 0.09 |
| 9-pixel | none | **76.74/6.41** | 78.34/1.73 | 77.93/7.44 | 80.97/2 | 76.67/2.88 | 78.59/5.35 | 75.47/59.62 | 82.49/9.05 | 83.16/25.06 | 82.6/2.54 |
| | Multikrum | 70.86/192.91 | 72.9/12.85 | **70.77/24.67** | **68.14/38.33** | 76.62/7.39 | 78.98/11.34 | 78.61/108.58 | **65.93/6.91** | 80.25/41.75 | **72.16/45.27** |
| | Bulyan | 71.85/61.15 | **61.27/15.79** | 75.92/0.43 | 73.17/72.43 | **63.72/25.07** | 74.72/66.57 | 71.16/88.44 | **58.54/253.43** | 81.17/70.54 | **65.01/70.88** |
| 1-pixel | none | **50.56/11.07** | 54.89/1.41 | 79.5/1.59 | 55.33/2.77 | **50.32/2** | 84.11/25.28 | 50.09/25.79 | **48.71/3.67** | 62.23/49.89 | 54.86/59.56 |
| | Multikrum | **55.01/0.28** | 51.37/21.02 | 68.21/94.76 | **47.59/8.54** | 74.59/5.27 | 56.96/31.38 | 54.45/28.54 | **57.06/0.05** | 41.25/5.96 | 73.25/41.74 |
| | Bulyan | 61.46/20.4 | **56.7/3.86** | 78.9/0.23 | 77.83/15.99 | **37.91/22.21** | 58.86/3.5 | 64.63/73.09 | **55.57/56.07** | 52.87/92.5 | 62.96/28.86 |

where

$$D = E||\gamma Regu||_2^2, \tag{24}$$

$$C = \frac{N-K}{N-1}\frac{4}{K}E^2(d^2q^2 + G^2 + D^2), \tag{25}$$

$$M = \sum_{k=1}^{N} p_k^2(d^2q^2 + \sigma_k^2 + D^2) + 6l\Gamma + 8(I-1)^2(d^2q^2 + G^2 + D^2). \tag{26}$$

Compared with LeadFL, the EA-PS method has a larger convergence upper bound, because of the parameter constraint and its regulation in Equation (24). **A higher tolerance to perturbations is achieved at the cost of a reduction in the convergence speed. It is noteworthy that in experiments, we set the same number of epochs as in LeadFL and still achieved better and more robust results.** The specific proof is presented in Appendix A.3.

### 5.3. Robustness Analysis

In this subsection, we analyze the robustness of EA-PS using the certified radius framework proposed by (Panda et al., 2021) for the case of periodic attacks. We provide definitions and assumptions in Appendix A.1.
Minimizing the certified radius is an upper bound that minimizes the distance between the poisoned model and the benign model, which improves the robustness of the model (Xie et al., 2021). We consider a general threat model where the number of malicious clients in each round of attacks is random. Then, the certified radius of EA-PS combined with any given server-side defense is derived as:

**Certified Radius**: *Let $f$ be a $c$-coordinatewise-Lipschitz protocol on a dataset $\Omega$. Then $R(\rho) = \Lambda(T)(1+dc)^{\Lambda(T)}\rho$ is a certified radius for $f$, where $\Lambda(t)$ is the cumulative learning rate $\Lambda(t) = \sum_{t=0}^{T-1}\eta_t$, $d$ is the dimension of model parameters.*

**Theorem 5.2.** *(Certified Radius): Let Assumption A.9 hold. The certified radius of the threat model is*

$$R(\rho) = (1+dc)^{\sum_{t\in\Phi_T}\eta_t}\rho$$
$$(|\prod_{t\in\Gamma_T}\sum_{k\in S_t^*}p^k(\frac{N}{|S_t^*|}A_t)| + |\Phi_T|\sum_{t\in\Phi_T}\eta_t),$$

*where $\Phi_T$ is the set of communication rounds that server-side defenses cannot filter out all malicious updates. $\Gamma_T$ is the set of communication rounds that server-side defenses filter out all malicious updates. $S_t^*$ is a set of clients whose updates are not filtered out by the server-side defense in round $t$. $K_m^t$ is the number of malicious clients selected in round $t$. $g_{atk}$ is the probability that the server-side defense filters out all malicious updates versus the number of malicious clients selected in a communication round. $|\Phi_T|$ and $|S_t^*|$ are the cardinality of the set $\Phi_T$ and $S_t^*$, where $E[|\Phi_T|] = \sum_{t=0}^{T-1}g_{atk}(K_m^t)$.*

From Theorem 4.1., we have a smaller coefficient of attack impact $A$ compared with LeadFL in the same environment.

So, we have a smaller certified radius. The specific proof is presented in Appendix A.4.

## 6. Experiments

In our experiments, we evaluate the EA-PS method with multiple server-side defense methods on FashionMNIST and CIFAR10 datasets under both IID and non-IID settings. We perform all baseline experiments based on the source code of LeadFL (Zhu et al., 2023)[1]. Our code can be found at https://anonymous.4open.science/r/EA-SP-6BC9.

Our goal is to maintain main task accuracy and at the same time to achieve better and more stable backdoor defense performance. So, **main task accuracy(MA)**, **backdoor accuracy(BA)** and its **max**, **min**, **variance** are used in this paper. We use the 1/9-pixel pattern backdoor attacks from (Bagdasaryan & Shmatikov, 2020). We use the settings that achieved the best results in the original papers.

We use CMA & CTMA (Yin et al., 2018), Multi-Krum (Mhamdi et al., 2018a) and Bulyan (Blanchard et al., 2017) as server-side defense methods. For client-side defense methods, we choose FL-WBC (Sun et al., 2021), LDP (Nas), LeadFL(Zhu et al., 2023), LeadFL with our parameter constraint strategy (noted as LeadFL$^+$) and EA-PS$^-$ as baseline methods.

### 6.1. Effectiveness of proposed **methods**

Table 1 shows the defense results against 9-pixel and 1-pixel attacks under IID and non-IID distributions on FashionMNIST and CIFAR10 datasets. For the FashionMNIST dataset, LeadFL$^+$, EA-PS$^-$ and EA-PS methods outperform LeadFL in backdoor accuracy and its variance, which indicates that our methods can effectively defend against backdoor attacks. By comparing LeadFL with LeadFL$^+$ and EA-PS$^-$ with EA-PS, we observe that the parameter constraint strategy can improve the performance of BA by up to 8.55% and increase variance by up to 20% to ensure the effect and its stability. Comparing LeadFL with EA-PS$^-$ and LeadFL$^+$ with EA-PS, we find that the defense effect of EA-PS$^-$ and EA-PS is significantly higher than that of LeadFL and LeadFL$^+$, which illustrate that the proposed new objective function can guarantee a more effective defense performance.

Meanwhile, we find that LeadFL$^+$, EA-PS$^-$ and EA-PS methods have a more balanced performance on the CIFAR10 dataset compared with the FashionMNIST dataset in the face of different server-side methods. By comparing LeadFL with LeadFL$^+$, and EA-PS$^-$ with EA-PS, We find that the BA is improved by up to 14.9%, and the variance is improved by up to 40%. This also illustrates the ef-

---

[1]https://github.com/CarlosChu-c/LeadFL.

fectiveness of our parameter constraint strategy. Comparing LeadFL with EA-PS$^-$ and LeadFL$^+$ with EA-PS, we get the same conclusion that the proposed new objective function can guarantee a more effective defense performance.

Finally, comparing the improvement of LeadFL$^+$ and EA-PS$^-$ with respect to LeadFL individually, we get two observations. 1) EA-PS$^-$ has a significantly higher improvement in BA than LeadFL$^+$, which indicates the proposed objective function has a better defense capability than the proposed parameter constraint strategy. 2) EA-PS$^-$ has a far less improvement of stability than LeadFL$^+$, which indicates the proposed parameter constraint strategy has a significant capability to maintain defense stability.

In addition, we conducted comparative experiments on None (FedAvg), LDP (Nas), and FL-WBC (Sun et al., 2021) methods to prove the effectiveness of our proposed methods. The specific BA, variance, upper and lower bounds, and other information are shown in Table 6-13 of Appendix C.

### 6.2. Impact of dynamic coefficient of the spatial mapping $\beta$

Table 2 shows how the performance of EA-PS varies with the impact of the dynamic coefficient of the spatial mapping $\beta$ under 1/9-pixel attacks with IID and non-IID distributions on the CIFAR10 dataset. Firstly, comparing the performance on different $\beta$, we can observe that EA-PS method is more stable under 9-pixel attacks than under 1-pixel attacks, which is because 9-pixel attacks are less aggressive than 1-pixel attacks. Then, we observe that under the influence of data heterogeneity, EA-PS method has a more stable BA performance under IID distribution than under non-IID distribution. Meanwhile, comparing the variance of BA under different distributions and under different attacks, we observe that attacks have a greater impact on the defense performance stability of EA-PS method. We also summarized the mean variance of BA under different $\beta$, and got the result that the difference between the mean variance under different $\beta$ does not exceed 10%, which indicates that the dynamic coefficient of the spatial mapping in parameter constraint strategy can effectively ensure the stability of the defense performance.

In addition, we also conduct comparative experiments on the FashionMNIST dataset in Table 14 of Appendix C, which has a more stable performance under different $\beta$ compared with the CIFAR10 dataset.

### 6.3. Impact of constraint rate $\alpha$

Table 3 shows how the defense performance varies with $\alpha$ under 1/9 pixel attacks with IID and non-IID distributions on the CIFAR10 dataset. By averaging BA and its VAR for each $\alpha$ setting, we find that when $\alpha$ tends to 0.1, BA is the

*Table 3.* Comparison of different $\alpha$ on CIFAR10 with IID/non-IID settings under 1/9-pixel backdoor attack.

| $\alpha$ (mean(%)/var($10^{-4}$)) | server-defense | 0.1 | 0.2 | 0.3 | 0.4 | 0.5 | 0.6 | 0.7 | 0.8 | 0.9 |
|---|---|---|---|---|---|---|---|---|---|---|
| IID, 9-pixel | MultiKrum | 65.34/20.48 | 72.99/7.15 | 81.1/1.46 | 82.8/0.55 | 85.71/0.29 | 81.92/0.1 | 83.17/0.25 | 82.35/8.48 | 81.93/6.06 |
| | Bulyan | 71.85/61.15 | 81.31/43.34 | 80.4/25.6 | 81.78/20.34 | 82.49/18.18 | 83.2/0.45 | 83.19/12.16 | 82.81/8.32 | 82.62/9.85 |
| IID, 1-pixel | MultiKrum | 55.01/0.28 | 41.63/30.42 | 59.1/8.12 | 67.15/12.04 | 73.24/2.02 | 70.78/9.41 | 66.01/5.5 | 59.54/5.58 | 66.48/13.59 |
| | Bulyan | 61.46/20.4 | 59.44/15.97 | 63.58/18.57 | 74.05/6.16 | 71.56/25.83 | 70.64/25.89 | 72.48/12.82 | 59.36/17.53 | 69.78/12.85 |
| non-IID, 9-pixel | MultiKrum | 65.46/21.76 | 79.87/5.45 | 85.55/1.02 | 86.51/0.1 | 83.89/0.75 | 82.91/0.7 | 84.87/0.3 | 84.28/8.79 | 82.79/3.37 |
| | Bulyan | 61.23/10.32 | 66.63/2.93 | 65.89/3.5 | 63.61/2.05 | 64.28/3.61 | 63.18/4.42 | 64.83/17.11 | 62.24/0.36 | 64.16/16.45 |
| non-IID, 1-pixel | MultiKrum | 61.56/7.73 | 50.71/1.8 | 72.09/27.12 | 72.18/22.03 | 64.44/20.79 | 75.7/24.06 | 67.53/14.27 | 73.09/17.07 | 66.3/12.59 |
| | Bulyan | 60.34/41.34 | 61.8/15.73 | 79.35/16.06 | 66.04/37.11 | 73.12/37.14 | 71.62/26.79 | 68.64/31.41 | 73.16/46.95 | 66.05/47.85 |

*Table 4.* Comparison of different linear ratio in $\lambda$ on CIFAR10 with IID/non-IID settings under 1/9-pixel backdoor attack.

| (mean(%)/var($10^{-4}$)) | | IID | | | | | non-IID | | | | |
|---|---|---|---|---|---|---|---|---|---|---|---|
| linear ratio in $\lambda$ | | 0.1 | 0.3 | 0.5 | 0.7 | 0.9 | 0.1 | 0.3 | 0.5 | 0.7 | 0.9 |
| 9-pixel | MultiKrum | 75.96/29.34 | 57.25/30.96 | 82.31/22.03 | 62.61/101 | 78.31/5.83 | 81.68/28.58 | 80.97/0.53 | 60.52/54.78 | 63.23/226.15 | 77.72/10.92 |
| | Bulyan | 59.66/88.28 | 78.35/35.77 | 44.1/32.43 | 40.49/68.72 | 76.76/10.94 | 78.08/4.96 | 83.51/1.99 | 61.23/10.32 | 81.22/3.02 | 65.42/11.46 |
| 1-pixel | MultiKrum | 73.3/19.79 | 43.71/8.17 | 55.44/13.76 | 72.53/25.71 | 57.55/0.21 | 59.85/12.8 | 67.02/83.97 | 60.23/3.22 | 51.58/16.64 | 51.59/6.72 |
| | Bulyan | 58.24/1.96 | 63.64/14.1 | 58.5/4.36 | 51.88/42.06 | 61.95/13.18 | 60.41/291.3 | 70.74/37.66 | 59.52/10.74 | 64.1/19 | 61.57/89.74 |

*Table 5.* Comparison of different $\gamma$ on CIFAR10 with IID/non-IID settings under 1/9-pixel backdoor attack.

| mean%/var$10^{-4}$ | | IID | | | | | non-IID | | | | |
|---|---|---|---|---|---|---|---|---|---|---|---|
| $\gamma$ | | 0.01 | 0.03 | 0.05 | 0.07 | 0.09 | 0.01 | 0.03 | 0.05 | 0.07 | 0.09 |
| 9-pixel | MultiKrum | 75.96/29.34 | 48.1/94.08 | 71.39/28.97 | 63.43/52.8 | 74.04/23.8 | 81.82/1.48 | 61.77/11.28 | 67.14/41.38 | 62.74/36.28 | 67.52/0.59 |
| | Bulyan | 84.04/40.54 | 78.2/1.25 | 75.76/1.59 | 54.88/69.25 | 79.04/5.47 | 79.92/41.5 | 75.08/22.06 | 66.31/2.95 | 22.06/11.38 | 62.5/29.81 |
| 1-pixel | MultiKrum | 78.92/6.89 | 55.14/1.83 | 48.54/17.69 | 78.2/4.11 | 78.8/39.37 | 59.85/12.8 | 51.03/0.04 | 57.22/10 | 43.1/40.95 | 66.8/14.91 |
| | Bulyan | 60.62/0.5 | 81.59/10.61 | 53.2/0.49 | 33.1/125.1 | 52.14/14.87 | 65.1/12.87 | 58.24/71.36 | 50.08/13.14 | 63.72/12.07 | 65.22/20.18 |

smallest but the variance is the largest. When $\alpha$ increases, BA increases but the variance decreases. But when $\alpha$ tends to 0.9, it is an exception that the variance becomes larger. The most suitable value of $\alpha$ is between 0.2 and 0.4, where the optimal variance is as low as $12.68 \times 10^{-4}$. This is because the parameter constraint weight increases with the value of $\alpha$, which makes the loss more biased to ensure stability. When the parameter constraint weight is too large, the variance increases because of the changed optimization space. This illustrates that the proposed objective function guarantees a smaller BA, while parameter constraint ensures the stability of the effect.

In addition, we also conducted comparative experiments on the FashionMNIST dataset and obtained the same conclusions (see Table 15 of Appendix C).

### 6.4. Impact of linear ratio in $\lambda$

Table 4 shows the effect of changing $\lambda$ under 1/9-pixel attacks with IID and non-IID distributions on the CIFAR10 dataset. Firstly, we observe that the BA performance of our method is more stable on 1-pixel attacks than on 9-pixel attacks. Meanwhile, we also observe that there is no significant difference on both distributions in the BA stability of our method. Then, by averaging the variance of each $\lambda$, we find that the defense performance is more stable when $\lambda$ is 0.5, where the optimal variance is as low as $18.96 \times 10^{-4}$. This is because the history information of $A_t$ and $A_{t-1}$ are taken into account with equal consideration.

In addition, we also conducted comparative experiments on the FashionMNIST dataset and obtained the same results (see Table 16 of Appendix C).

### 6.5. Impact of regulation rate $\gamma$

Table 5 shows the effect of changing the regulation rate $\gamma$ under 1/9-pixel attacks with IID and non-IID distributions on the CIFAR10 dataset. Three observations are summarized as follows: 1) The defense performance is more stable under the non-IID distribution than IID distribution varying with $\gamma$; 2) There is no significant difference in the BA stability of our method under two attacks; 3) Our method get best results when $\gamma$ centered around 0.05, with an average BA of 61.21% and an average variance of $14.53 \times 10^{-4}$.

In addition, we also conducted comparative experiments on the FashionMNIST dataset and obtained similar results (see Table 17 of Appendix C).

## 7. Conclusion

To defend against persistent adaptive attacks with long-lasting attack effects, we propose a client-side defense method, EA-PS, which can be effectively combined with server-side methods to guarantee robust and stable performance. Benefiting from minimizing the impact of attacks and the constraint of the perturbation range of local parameters, EA-PS method effectively thwarts backdoor poisoning attacks with stable performance. To theoretically guarantee the performance and robustness of EA-PS, we prove that our methods have a lower upper bound, a smaller certified radius, and a larger convergence upper bound. Evaluated on FashionMNIST and CIFAR10 combined with different server-side defense methods under both IID and non-IID data distributions, EA-PS achieves lower attack success rates by up to 14.9% and more stable defense performance with smaller variance by up to 40% compared with other client-side defense methods.

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

## Appendix for
## "EA−PS: Estimated Attack Effectiveness based Poisoning Defense in Federated Learning under Parameter Constraint Strategy"

In the appendix of this paper, we provide further details:

- In Appendix A, we first show that EA−PS has a smaller optimization upper bound (A.2), which guarantees better optimization results for this method. Secondly, it is proved that EA−PS has a larger radius of convergence (A.3), which ensures that the backdoor defense method has a larger effective range. Finally, it is proved that EA−PS method has a smaller robust certified radius (A.4), which ensures that the backdoor defense method has strong robustness.

- In Appendix B, we present the details of our experiments, including the dataset, the server-side & client-side defense methods, and the detailed experimental configuration, such as client selection and rounds, training details, model architectures, and evaluation metrics.

- In Appendix C, we show the detailed results (including main task accuracy, backdoor accuracy and their MAX, MIN and VAR values) of the baselines and `proposed methods` (including LeadFL$^+$, EA−PS$^-$, and EA−PS ) against 1/9 pixel attacks under IID and non-IID distributions on the FashionMNIST and CIFAR10 datasets.

- In Appendix D, we discuss the problem of dataset selection, baseline selection, performance improvement, and heterogeneity applicability of the proposed method in this paper.

## A.Proofs

### A.1 Assumptions and Definitions

**Assumption A.1** (Smoothness). $L$ is $\ell - smooth$ if $\forall x, y \in \Re^d$

$$L(x) - L(y) + (x - y)^T \bigtriangledown L(x) \leq \frac{\ell}{2} ||x - y||_2^2.$$

**Assumption A.2** (Convex). $L$ is $\mu - strongly$ convex if $\forall x, y \in \Re^d$

$$L(x) - L(y) + (x - y)^T \bigtriangledown L(y) \geq \frac{\mu}{2} ||x - y||_2^2.$$

**Assumption A.3** (Bound of Variance). Let $\xi_t^k$ be sampled from the $k$-th device's local data uniformly at random. The variance of the stochastic gradient in each device is bounded: $E|| \bigtriangledown L^k(\theta_t^k, \xi_t^k) - \bigtriangledown L^k(\theta_t^k)||^2 \leq \sigma_k^2$ for $k = 1, ..., N$.

**Assumption A.4** (Bound of Norm). The expected squared norm of stochastic gradients is uniformly bounded, i.e., $E|| \bigtriangledown L^k(\theta_{t,e}^k, \xi_{t,e}^k)|| \leq G^2$ for all $k = 1, ..., N$, $e = 0, ..., E - 1$ and $t = 0, ..., T - 1$.

**Assumption A.5** (Selection of Clients). Assume $S_t$ contains a subset of K indices uniformly sampled from [N] without replacement. Assume the data is balanced in the sense that $p_1 = ... = p_N = \frac{1}{N}$. The aggregation step of FedAvg performs $\theta_t \longleftarrow \frac{N}{K} \sum_{k \in S_t} p_k \theta_t^k$.

**Definition A.6** (Loss of clients). Denote $L^*$ and $L_k^*$ as the minimum value of $L$ and $L_k$, where $L$ is the loss of a model trained on the combination of datasets from all the clients and $L_k$ is the loss of a model trained on the dataset of client $k$. we can set $\Gamma = L^* - \sum_{k=1}^N p_k L_k^*$, which can quantify the degree of non-IID. If the data are IID, then $\Gamma$ goes to zero as the number of samples grows. If the data are non-IID, the $\Gamma$ is non-zero, and its magnitude reflects the heterogeneity of the data distribution.

**Definition A.7** (Poisoning Attack). For a protocol $f = (\mathcal{G}, \mathcal{A}, \eta)$ we define the set of poisoned protocols $F(\rho)$ to be all protocols $f^* = (\mathcal{G}^*, \mathcal{A}, \eta)$ that are exactly the same as $f$ except that the gradient oracle $\mathcal{G}^*$ is a $\rho - corrupted$ version of $\mathcal{G}$. That is, for any round $t$ and any model $\theta_t$ and any dataset $D$ we have $\mathcal{G}^*(\theta_t, D) = \mathcal{G}(\theta_t, D) + \epsilon$ for some $\epsilon$ with $||\epsilon||_1 \leq \rho$.

**Definition A.8** (Certified Radius). Let $f$ be a protocol and $f^* \in F(\rho)$ be a poisoned version of the same protocol. Let $\theta_T, \theta_T^*$ be the benign and poisoned final outputs of the above protocols. We call $R$ a certified radius for $f$ if $\forall f^* \in F(\rho); R(\rho) \geq |\theta_T - \theta_T^*|_1$.

**Assumption A.9** (Coordinate-wise Lipschitz). The protocol $f(\mathcal{G}, \mathcal{A}, \eta)$ is $c$-coordinate-wise Lipschitz if for any round $t \in [T]$, models $\theta_t, \theta_t^* \in \mathcal{M}$, and a dataset $D$ we have that the outputs of the gradient oracle on any coordinate can't drift too much farther apart. Specifically, for any coordinate index $i \in [d]$

$$|\mathcal{G}(\theta_t^*, D)[i] - \mathcal{G}(\theta_t, D)[i]| \leq c|\theta_t^* - \theta_t|_1.$$

**A.2 Proof of Theorem 4.1**

**Theorem 4.1.** *Minimizing $A_t - A_{t-1}$ yields a smaller optimization upper bound than minimizing $A_{\hat{t}}$, where $A_{\hat{t}}$ is the coefficient of attack impact in LeadFL.*

*Proof.*

From the definition of $A_t$, we can get Equations (27) as follows:

$$A_t = I - (P_t - P_{t-1}) + \Delta_t. \tag{27}$$

According to Equation (27), we can obtain:

$$A_t - A_{t-1} = (\Delta_t - \Delta_{t-1}) - (P_t - 2P_{t-1} + P_{t-2}), \tag{28}$$

where we assume that the lower bound on the difference between $A_t - A_{t-1}$ and $0$ is $\epsilon$. For convenience, we consider this difference to be $\epsilon$.

According to Equation (28), we can obtain the recursion formula as follows:

$$\begin{cases} P_2 = \Delta_2 - \Delta_1 + 2P_1 - P_0 + \varepsilon \\ P_3 = \Delta_3 - \Delta_2 + 2P_2 - P_1 + \varepsilon \\ \qquad\qquad \vdots \\ P_t = \Delta_t - \Delta_{t-1} + 2P_{t-1} - P_{t-2} + \varepsilon. \end{cases} \tag{29}$$

The recursive formula can be obtained from Equation (29) as follows:

$$\begin{cases} P_3 = \Delta_3 + \Delta_2 - 2\Delta_1 + 3P_1 - 2P_0 + 3\varepsilon \\ P_4 = \Delta_4 + \Delta_3 + \Delta_2 - 3\Delta_1 + 4P_1 - 3P_0 + 6\varepsilon \\ \qquad\qquad \vdots \\ P_t = \sum_{i=1}^{t} \Delta_i + P_0 + [(t-1) + \varepsilon_{(P_{t-1})}]\varepsilon. \end{cases} \tag{30}$$

From the definition of $A_{\hat{t}}$, we can get Equations (31) as follows:

$$A_{\hat{t}} = I - (P_{\hat{t}} - P_{\hat{t}-1}) + \Delta_{\hat{t}}. \tag{31}$$

Similarly, assume that the lower bound on the difference between $A_{\hat{t}}$ and $0$ is also $\epsilon$.

According to Equation (31), we can obtain the recursion formula as follows:

$$\begin{cases} P_1 = P_0 + I + \Delta_1 + \varepsilon \\ P_2 = P_1 + I + \Delta_2 + \varepsilon \\ P_3 = P_2 + I + \Delta_3 + \varepsilon \\ \qquad\qquad \vdots \\ P_{\hat{t}} = P_{\hat{t}-1} + I + \Delta_{\hat{t}} + \varepsilon. \end{cases} \tag{32}$$

The recursive formula can be obtained from Equation (32) as follows:

$$\begin{cases} P_2 = P_0 + 2I + \Delta_1 + \Delta_2 + 2\varepsilon \\ P_3 = P_0 + 3I + \Delta_1 + \Delta_2 + \Delta_3 + 2\varepsilon \\ \qquad\qquad \vdots \\ P_{\hat{t}} = P_0 + \hat{t}I + \sum_{k=1}^{\hat{t}} \Delta_k + \hat{t}\varepsilon. \end{cases} \tag{33}$$

According to the Equation (28) and (30), we can obtain Equation (34) as follows:

$$A_t = I - (P_t - P_{t-1}) + \Delta_t = I + \Delta_1 - P_1 + P_0 - t\varepsilon. \tag{34}$$

According to the Equation (31) and (33), we can obtain Equation (35) as follows:

$$A_{\hat{t}} = I - (P_{\hat{t}} - P_{\hat{t}-1}) + \Delta_{\hat{t}} = I + \varepsilon. \tag{35}$$

We can obtain that $A_t$ has a lower upper bound than $A_{\hat{t}}$ as follows:

$$A_{\hat{t}} - A_t = (t+1)\varepsilon + P_1 - P_0 - \Delta_1 = (t+1)\varepsilon. \tag{36}$$

According to Equation (36), $A_t \leq A_{\hat{t}}$. Then we can get Theorem 4.1 that minimizing $A_t - A_{t-1}$ yields a smaller optimization upper bound than minimizing $A_{\hat{t}}$.

**A.3 Proof of Convergence Guarantee**

**Theorem 5.1. Convergence Guarantee:** *Let Assumptions A.2 to A.6 hold and $l, \mu, \sigma_k, G, K, N, \Gamma, F^*$ be defined therein and in Definition A.7. Choose $\kappa = \frac{l}{\mu}$, $\varphi = max(8\kappa, I)$ and the learning rate $\eta_t = \frac{2}{\mu(\varphi+t)}$. Then we have the following bound for* EA-PS:

$$E[F(\theta_T)] - F^* \leq \frac{\kappa}{\varphi + T - 1}(\frac{2(M+C)}{\mu} + \frac{\mu\varphi}{2}E[||\theta_0 - \theta^*||^2]), \tag{37}$$

*where*

$$D = E||\gamma Regu||_2^2, \tag{38}$$

$$C = \frac{N-K}{N-1}\frac{4}{K}E^2(d^2q^2 + G^2 + D^2), \tag{39}$$

$$M = \sum_{k=1}^{N} p_k^2(d^2q^2 + \sigma_k^2 + D^2) + 6l\Gamma + 8(I-1)^2(d^2q^2 + G^2 + D^2), \tag{40}$$

*Proof.*

The expected distance between the gradients before and after regularization can be bounded.

$$E||\bigtriangledown L^{'}(\theta_{t,i}^k, \xi_{t,i}^k) - \bigtriangledown L(\theta_{t,i}^k, \xi_{t,i}^k)||_2^2 \tag{41}$$

$$= E||clip(A_t, q) + \gamma Regu||_2^2 \tag{42}$$

$$\leq E||clip(A_t, q)||_2^2 + E||\gamma Regu||_2^2 \tag{43}$$

$$\leq d^2q^2 + D^2. \tag{44}$$

Using the bounds above and Assumption A.3, we can derive new bounds for the variance of modified gradient $E||\bigtriangledown L^{'}(\theta_{t,i}^k, \xi_{t,i}^k) - \bigtriangledown L(\theta_{t,i}^k)||^2$

$$E||\bigtriangledown L^{'}(\theta_{t,i}^k, \xi_{t,i}^k) - \bigtriangledown L(\theta_{t,i}^k)||^2 \tag{45}$$

$$\leq E||\bigtriangledown L^{'}(\theta_{t,i}^k, \xi_{t,i}^k) - \bigtriangledown L(\theta_{t,i}^k, \xi_{t,i}^k)||^2 \tag{46}$$

$$+ E||\bigtriangledown L(\theta_{t,i}^k, \xi_{t,i}^k) - \bigtriangledown L(\theta_{t,i}^k)||^2 \tag{47}$$

$$\leq d^2q^2 + \sigma_k^2 + D^2, \tag{48}$$

where we use the triangle inequality. Similarly, we can also derive bounds for the expected squared norm of modified gradients using Assumption A.4.

$$E||\bigtriangledown L^{'}(\theta_{t,i}^k, \xi_{t,i}^k)||^2 \tag{49}$$

$$\leq E||\bigtriangledown L^{'}(\theta_{t,i}^k, \xi_{t,i}^k) - \bigtriangledown L(\theta_{t,i}^k, \xi_{t,i}^k)||_2^2 \tag{50}$$

$$+ E||\bigtriangledown L(\theta_{t,i}^k, \xi_{t,i}^k)||^2 \tag{51}$$

$$\leq d^2q^2 + G^2 + D^2. \tag{52}$$

Applying the bounds for the variance and the expected squared norm of modified gradients after applying EA-PS, we can derive our convergence guarantee from Theorem 5.1 in (Li et al., 2019b) by replacing these bounds. Compared with LeadFL, the EA-PS method has a larger convergence upper bound, because of the parameter constraint and its regulation in Equation (38).

### A.4 Proof of Certified Radius of the Threat Model

Our paper uses the same Poisoning Attack Definition (Definition A.7) and Coordinate-wise Lipschitz Assumption (Assumption A.9) as LeadFL.

**Theorem 5.2. Certified Radius of the Threat Model:** *EA-PS has a smaller upper bound and its certified radius is also smaller than LeadFL.*

*Proof.*

From the proof of the certified radius in LeadFL, it is known that under the assumption of unification, EA-PS has a smaller upper bound and its certified radius is also smaller than LeadFL. Based on the definition of model updates, we use the triangle inequality to get the following inequality between $|\theta_t - \theta_t^*|$ and $|\theta_{t-1} - \theta_{t-1}^*|$ when the system is attacked in round $t-1$.

$$|\theta_t - \theta_t^*| = |\theta_{t-1} - \eta_t \mu_t - \theta_{t-1}^* + \eta_t \hat{\mu}_t| \leq |\theta_{t-1} - \theta_{t-1}^*| + \eta_t |\mu_t - \hat{\mu}_t|. \tag{53}$$

Using the triangle inequality again, we can get:

$$|\mu_t - \hat{\mu}_t| = |\mu_t - \mu_t^* + \mu_t^* - \hat{\mu}_t| \leq |\mu_t - \mu_t^*| + |\mu_t^* - \hat{\mu}_t|. \tag{54}$$

According to Definition A.7 and coordinate-wise Lipshitz in Assumption A.9:

$$|\mu_t - \hat{\mu}_t| \leq |\mu_t - \mu_t^*| + |\mu_t^* - \hat{\mu}_t| = dc|\theta_t - \theta_t^*| + \rho. \tag{55}$$

By plugging the above equation into Equation (53), we get:

$$|\theta_t - \theta_t^*| \leq |\theta_{t-1} - \theta_{t-1}^*| + \eta_t(dc|\theta_t - \theta_t^*| + \rho) = (1 + dc\eta_t)|\theta_t - \theta_t^*| + \rho\eta_t. \tag{56}$$

According to Bernoulli's inequality, we have:

$$|\theta_t - \theta_t^*| \leq (1 + dc)^{\eta_t}|\theta_t - \theta_t^*| + \rho\eta_t. \tag{57}$$

Now we get the inequality between $|\theta_t - \theta_t^*|$ and $|\theta_{t-1} - \theta_{t-1}^*|$ when the system is attacked in round $t-1$. Since we introduced server-side defense, we obtain the Equation (58) from the Equation (4):

$$\theta_t - \theta_t^* = \sum_{k \in S_t^*} p^k \left| \prod_{i=0}^{I-1} \left( \frac{N}{|S_t^*|} A_t \right) \right| (\theta_{t-1} - \theta_{t-1}^*). \tag{58}$$

Then we get the following relationship between $|\theta_t - \theta_t^*|$ and $|\theta_{t-1} - \theta_{t-1}^*|$ when server-side defense filters out all malicious updates in round $t-1$.

$$|\theta_t - \theta_t^*| \leq \sum_{k \in S_t^*} p^k \left| \prod_{i=0}^{I-1} \left( \frac{N}{|S_t^*|} A_t \right) \right| |\theta_{t-1} - \theta_{t-1}^*|. \tag{59}$$

Finally, we can use Equations (57) and (59) to prove Theorem 5.2 by induction hypothesis:

$$R(\rho) = (1 + dc)^{\sum_{t \in \Phi_T} \eta_t} \rho(| \prod_{t \in \Gamma_T} \sum_{k \in S_t^*} p^k (\frac{N}{|S_t^*|} A_t)| + |\Phi_T| \sum_{t \in \Phi_T} \eta_t). \tag{60}$$

From Theorem 4.1., we have a smaller coefficient of attack impact $A$ compared with LeadFL in the same environment. So, we have a smaller certified radius.

## B. Experiments Detail

### B.1. Datasets

We conduct experiments on FashionMNIST and CIFAR10. In the case of FashionMNIST, every one of the 100 clients is allocated 600 images from a total of 60,000 images. As for CIFAR10, each client obtains 500 images out of the 50,000 available images.

In the IID setting, samples are uniformly distributed to clients. In the non-IID setting, we deploy the limited label strategy (McMahan et al., 2016) that is also used for the evaluation of LeadFL in FashionMNIST and CIFAR10: Of the 10 classes in each of the two datasets, each client is assigned 5 random classes. They are then assigned an equal number of randomly selected samples from each of their classes. The clients' datasets are selected independently.

For the regularization term, we tune the parameters of the Dirichlet distribution in the non-IID case using hyper-parameters $\alpha$. Here we set $\alpha = 0.4$ in FashionMNIST and 0.25 in CIFAR10.

### B.2. Server-side & Client-side Defenses

We use CMA&CTMA (Yin et al., 2018), Multi-Krum (Mhamdi et al., 2018a) and Bulyan (Blanchard et al., 2017) as server-side defenses.

For client-side defenses, we choose FL-WBC (Sun et al., 2021), LDP (Nas), LeadFL, LeadFL with our parameter constraint strategy (noted as `LeadFL`$^+$) and `EA-PS`$^-$ as the baseline. For FL-WBC (Sun et al., 2021) and LDP (Nas) defenses, we apply Laplace noise with $mean = 0$ and $std = 0.2$ as in the original papers. For LeadFL and `EA-PS`$^-$, we set the clipping norm $q = 0.2$. For `LeadFL`$^+$, we set the dynamic coefficient of the spatial mapping $\beta = 0.05$, regularization rate $\alpha = 0.1$ and linear ratio of $\lambda$ to 0.5.

### B.3. Configurations

**Client Selection and Rounds.** There are 100 clients in total, of which 25 are malicious. There are 80 global rounds and 10 local rounds. The server selects 10 clients per global round. For most experiments, the selection is random but consistent over experiments, i.e., for two experiments, the clients selected in round t are the same to enable comparison between the different settings.

**Training Details.** For training, we set local epoch $E$ as 1 and batch size $BS$ as 32. We apply SGD optimizer and set the learning rate $\eta$ to 0.01. Up to five clients are selected as malicious clients in each round of 80 communication rounds. Our hyper-parameters are **(1)** $\beta$ is the initial value of adaptive parameter constraint, set to 0.01. **(2)** $\alpha$ is the ratio of adaptive parameter constraint to `EA-PS`$^-$, set to 0.1. **(3)** $\lambda$ is the ratio of linear decision rules, set to 0.5. **(4)** $\gamma$ is the ratio of regulation to control the influence degree of $\beta$, set to 0.01.

**Model Architectures.** We adopt the same model architecture as LeadFL (Zhu et al., 2023) on the FashionMNIST and CIFAR10 with convolutional layers and fully-connected layers.

**Evaluation Metrics.** Our goal is to maintain main task accuracy and at the same time achieve better and more stable backdoor defense performance. So, **Main Task Accuracy(MA)**, **Backdoor Accuracy(BA)** and its **MAX**, **MIN**, **Variance** are used in this paper. (1)**Main Task Accuracy(MA)**: We measure the main task accuracy using the accuracy of the global model on the benign test set of the main task. As in other works, we consider the maximum accuracy achieved during training. (2)**Backdoor Accuracy(BA)**: Backdoor accuracy measures how successful an attacker is in integrating the backdoor into the model. We measure the accuracy of the backdoor as the percentage of samples with triggers that are classified as attacker intent. We find that the backdoor accuracy does not converge in our experiments, so we consider the average backdoor accuracy. And because the difference in backdoor accuracy is large in each round, we use the average backdoor accuracy in our experiments. At the same time, we also give the **MAX** and **MIN** of the experimental results, which represent the defense effect interval of the `EA-PS` method. (3)**Backdoor Accuracy Variance**: We measure the backdoor accuracy variance to represent the stability of the defense effect.

## C. Additional Results

Table 6 shows the results of different client-side defense methods combined with different server-side defense methods in the **FashionMNIST** dataset under **IID** distribution in the case of **9-pixel** attacks.

Table 7 shows the results of different client-side defense methods combined with different server-side defense methods in the **FashionMNIST** dataset under **non-IID** distribution in the case of **9-pixel** attacks.

Table 8 shows the results of different client-side defense methods combined with different server-side defense methods in the **FashionMNIST** dataset under **IID** distribution in the case of **1-pixel** attacks.

Table 9 shows the results of different client-side defense methods combined with different server-side defense methods in the **FashionMNIST** dataset under **non-IID** distribution in the case of **1-pixel** attacks.

Table 10 shows the results of different client-side defense methods combined with different server-side defense methods in the **CIFAR10** dataset under **IID** distribution in the case of **9-pixel** attacks.

Table 11 shows the results of different client-side defense methods combined with different server-side defense methods in the **CIFAR10** dataset under **non-IID** distribution in the case of **9-pixel** attacks.

Table 12 shows the results of different client-side defense methods combined with different server-side defense methods in the **CIFAR10** dataset under **IID** distribution in the case of **1-pixel** attacks.

Table 13 shows the results of different client-side defense methods combined with different server-side defense methods in the **CIFAR10** dataset under **non-IID** distribution in the case of **1-pixel** attacks.

Table 14 shows the experimental results of tuning the hyper-parameter $\beta$ on the **FashionMNIST** dataset under 1/9-pixel backdoor attacks.

Table 15 shows the experimental results of tuning the hyper-parameter $\alpha$ on the **FashionMNIST** dataset under 1/9-pixel backdoor attacks.

Table 16 shows the experimental results of tuning the hyper-parameter $\lambda$ on the **FashionMNIST** dataset under 1/9-pixel backdoor attacks.

Table 17 shows the experimental results of tuning $\gamma$ on **FashionMNIST** with IID/non-IID settings under 1/9-pixel backdoor attacks.

## D. Discussion

**Dataset Selection:** Due to the limitation of the paper's length, we only used two commonly used datasets for verification. Although our experimental results perform better on the FashionMNIST dataset, the CIFAR10 dataset is superior to the FashionMNIST dataset in terms of complexity, generalization, and scene diversity, we mainly adopt the experimental results of the CIFAR10 dataset in the analysis of the main text.

**Performance Improvement:** As can be seen from Table 13-17 in the appendix and the default experimental settings of this paper, there still exists much room for defense performance improvement of the proposed method through fine-tuning of hyperparameters. However, current experiments and theoretical analyses are sufficient to prove the superiority of the proposed method. Additionally, although our method has a larger convergence upper bound, we set the same number of epochs in the experiment for a fair comparison with methods such as LeadFL and WBC. To further improve defense performance, our method can appropriately increase the number of epochs.

**Baseline Selection:** In this paper, the proposed method is compared with the latest parameterized client-side methods. There are new developments in client-side methods from 2024 to 2025, but most of them are based on differential privacy, distillation learning, or malicious client detection, which means they are not comparable to the proposed method.

**Heterogeneity Applicability:** Our method is based on the client backdoor defense method, so our method can also perform backdoor defense under heterogeneity, but it needs to be adjusted adaptively according to the structure of different client networks.

*Table 6.* Comparison under 9-pixel pattern backdoor attack on IID FashionMNIST dataset

| Server-side Defense | Client-side Defense | MA | BA | VAR |
|---|---|---|---|---|
| None | None | 89.88 | 98.53 (+0.36 / -0.35) | 0.13 |
| | LDP | 88.36 | 90.81(+1.33 / -1.06) | 1.47 |
| | WBC | 88.23 | 90.30(+0.86 / -1.26) | 1.24 |
| | LeadFL | 87.42 | 93.22(+1.5/ -3.48) | 7.75 |
| | LeadFL$^+$ | 87.39 | 89.36(+1.73/-2.26) | 3.66 |
| | EA-PS$^-$ | 87.11 | 88.8(+2.35/ -1.22) | 1.86 |
| | EA-PS | 86.92 | 90.72(+1.8 / -2.13) | 3.02 |
| CMA | None | 89.79 | 96.65(+0.49 /-0.83) | 0.52 |
| | LDP | 87.03 | 96.66(+1.59 /-1.30) | 2.15 |
| | WBC | 87.19 | 96.78(+0.72 / -0.37) | 0.39 |
| | LeadFL | 87.72 | 95.2(+1.64/ -2.64) | 2.8 |
| | LeadFL$^+$ | 87.22 | 93.62(+1.55/-1.43) | 2.71 |
| | EA-PS$^-$ | 86.87 | 92.73(+3.3/ -2.39) | 6.91 |
| | EA-PS | 86.98 | 91.93(+2.36/ -1.97) | 4.23 |
| CTMA | None | 89.86 | 96.32(+0.38 / -0.61) | 0.29 |
| | LDP | 88.26 | 98.18(+0.11 / -0.06) | 0.01 |
| | WBC | 88.2 | 97.8(+0.18 /-0.17) | 0.03 |
| | LeadFL | 87.32 | 87.42(+4.55/ -6.17) | 15.04 |
| | LeadFL$^+$ | 86.82 | 83.65(+4.36/-5.62) | 16.77 |
| | EA-PS$^-$ | 86.67 | 79.38(+4.85/ -7.36) | 29.5 |
| | EA-PS | 86.76 | 84.65(+4.11/ -4.63) | 13.31 |
| multiKrum | None | 89.40 | 33.59(+37.15 /-18.95) | 1034.74 |
| | LDP | 86.78 | 76.41(+2.17 /-1.89) | 4.18 |
| | WBC | 86.83 | 77.52(+0.52 / -0.52) | 0.27 |
| | LeadFL | 86.72 | 32.82(+7.56 / -5.52) | 23.4 |
| | LeadFL$^+$ | 86.38 | 23.34(+5.15/-4.33) | 19.7 |
| | EA-PS$^-$ | 86.08 | 21.73(+6.7/ -9.75) | 52.13 |
| | EA-PS | 86.18 | 22.97(+3.29/ -3.16) | 7.95 |
| bulyan | None | 89.4 | 36.23(+46.58 /-23.93) | 1627.7 |
| | LDP | 85.88 | 74.34(+3.42/-2.74) | 9.84 |
| | WBC | 85.96 | 73.93(+4.16 /-6.82) | 35.29 |
| | LeadFL | 85.73 | 32.78(+10.54/ -13.99) | 99.34 |
| | LeadFL$^+$ | 85.44 | 22.57(+10.81/-9.39) | 84.13 |
| | EA-PS$^-$ | 85.03 | 19.7(+25.72/ -13.56) | 246.97 |
| | EA-PS | 85.77 | 19.59(+6.67/ -11.42) | 65.19 |

*Table 7.* Comparison under 9-pixel pattern backdoor attack on non-IID FashionMNIST dataset

| Server-side Defense | Client-side Defense | MA | BA | VAR |
|---|---|---|---|---|
| None | None | 89.69 | 97.97(+0.87 / -0.56) | 0.59 |
| | LDP | 87.88 | 88.24(+1.42 / -0.93) | 1.59 |
| | WBC | 88.05 | 89.4(+1.56 / -1.18) | 1.9 |
| | LeadFL | 87.21 | 92.43(+1.47/ -2.06) | 2.29 |
| | LeadFL$^+$ | 86.24 | 89.04(+1.56/-1.69) | 2.24 |
| | EA-PS$^-$ | 86.43 | 88.10(+3.54/ -8.27) | 13.98 |
| | EA-PS | 86.48 | 86.67(+2.77/ -0.676) | 2.15 |
| CMA | None | 89.67 | 95.07(+0.32 /-0.51) | 0.2 |
| | LDP | 86.86 | 95.18(+0.31 /-0.25) | 0.2 |
| | WBC | 86.3 | 96.1(+1.98 / -1.06) | 2.94 |
| | LeadFL | 87.51 | 91.93(+3.36/ -2.87) | 8.37 |
| | LeadFL$^+$ | 86.93 | 89.91(+4.06/-4.33) | 11.99 |
| | EA-PS$^-$ | 86.56 | 89.90(+4.66/ -4.71) | 18.91 |
| | EA-PS | 88.47 | 91.82(+1.11/ -1.08) | 0.68 |
| CTMA | None | 89.72 | 64.64(+4.3 / -3.77) | 16.54 |
| | LDP | 87.4 | 96.37(+1.36 / -1.19) | 3.25 |
| | WBC | 87.79 | 97.93(+0.29 /-0.14) | 0.06 |
| | LeadFL | 87.27 | 87.25(+7.67/ -3.88) | 24.10 |
| | LeadFL$^+$ | 86.62 | 88.01(+4.5/-4.53) | 18.09 |
| | EA-PS$^-$ | 86.04 | 80.88(+7.70/ -9.77) | 47.14 |
| | EA-PS | 86.48 | 81.28(+6.89/ -9.09) | 36.07 |
| multiKrum | None | 89.08 | 64.64(+4.3 /-3.77) | 16.54 |
| | LDP | 86.37 | 31.39(+3.11 / -3.15) | 9.81 |
| | WBC | 86.36 | 31.9(+4.14/-6.19) | 29.81 |
| | LeadFL | 85.94 | 17.38(+6.94 / -7.76) | 28.33 |
| | LeadFL$^+$ | 85.55 | 16.43(+6.65/-4.31) | 20.65 |
| | EA-PS$^-$ | 85.96 | 12.58(+4.96/ -5.11) | 17.4 |
| | EA-PS | 85.41 | 11.79(+2.52/ -2.95) | 4.76 |
| bulyan | None | 88.83 | 69.94(+2.45 / -3.39) | 9.17 |
| | LDP | 85.39 | 27.31(+4.99 / -4.61) | 23.11 |
| | WBC | 85.99 | 32.36(+3.67/-4.89) | 19.42 |
| | LeadFL | 85.3 | 15.14(+9.14/ -11.98) | 61.37 |
| | LeadFL$^+$ | 85.12 | 13.77(+8.39/-7.46) | 59.88 |
| | EA-PS$^-$ | 85.34 | 11.38(+16.85/ -7.63) | 93.83 |
| | EA-PS | 84.97 | 12.45(+6.2/ -3.53) | 22.72 |

*Table 8.* Comparison under 1-pixel pattern backdoor attack on IID FashionMNIST dataset

| Server-side Defense | Client-side Defense | MA | BA | VAR |
|---|---|---|---|---|
| None | None | 89.83 | 96.03(+0.33 / -0.26) | 0.08 |
| | LDP | 88.32 | 86.54(+1.27 / 0.68) | 0.77 |
| | WBC | 88.12 | 86.55(+0.4 / -0.33) | 0.09 |
| | LeadFL | 87.35 | 89.83(+2.1/ -1.73) | 2.49 |
| | LeadFL$^+$ | 86.99 | 86.42(+0.12/-0.86) | 0.68 |
| | EA-PS$^-$ | 86.6 | 86.24(+1.22/ -1.33) | 2.07 |
| | EA-PS | 86.75 | 88.76(+1.26 / -2.49) | 4.67 |
| CMA | None | 89.69 | 91.19(+1.66 /-2.39) | 3.54 |
| | LDP | 87.15 | 94.63(+1.7 /-1.30) | 1.56 |
| | WBC | 86.99 | 95.29(+0.83 / -0.67) | 0.39 |
| | LeadFL | 87.71 | 89.53(+1.54/ -1.94) | 2.76 |
| | LeadFL$^+$ | 87.33 | 86.24(+1.12/-1.66) | 2 |
| | EA-PS$^-$ | 86.86 | 85.48(+3.59/ -2.1) | 6.52 |
| | EA-PS | 86.87 | 84.84(+2.11/ -1.08) | 3.34 |
| CTMA | None | 89.85 | 92.47(+0.36 / -0.6) | 0.2 |
| | LDP | 88.08 | 96.32(+0.75 / -0.8) | 0.53 |
| | WBC | 87.88 | 87.71(+0.7 /-0.44) | 0.37 |
| | LeadFL | 87.17 | 87.71(+0.7/ -0.44) | 0.37 |
| | LeadFL$^+$ | 86.23 | 86.59(+1.32/-2.04) | 5.21 |
| | EA-PS$^-$ | 87.16 | 86.09(+1.53/ -1.54) | 4.69 |
| | EA-PS | 86.78 | 84.06(+4.13/ -8.11) | 49.33 |
| multiKrum | None | 89.62 | 22.6(+8.42 /-5.59) | 33.06 |
| | LDP | 86.89 | 71.45(+3.97 /-4.07) | 11.3 |
| | WBC | 86.7 | 72.18(+4.24 / -5.59) | 18.73 |
| | LeadFL | 86.77 | 60.5(+10.3 / -8.23) | 64.4 |
| | LeadFL$^+$ | 86.03 | 56(+11.06/-7.79) | 93.83 |
| | EA-PS$^-$ | 86.19 | 49.87(+7.45/ -4.67) | 42.56 |
| | EA-PS | 86.32 | 45.32(+4.5/ -3.91) | 17.94 |
| bulyan | None | 89.26 | 73.04(+7.39 /-12.98) | 83.48 |
| | LDP | 85.8 | 70.31(+2.56/-1.6) | 3.21 |
| | WBC | 85.63 | 68.39(+1.63 /-3.63) | 3.48 |
| | LeadFL | 85.63 | 58.72(+9.96/ -21.83) | 215.5 |
| | LeadFL$^+$ | 84.99 | 51.39(+5.28/-3.69) | 41.29 |
| | EA-PS$^-$ | 85.55 | 45.45(+10.56/ -22.91) | 237.83 |
| | EA-PS | 85.57 | 45.68(+9.51/ -15.77) | 189.3 |

*Table 9.* Comparison under 1-pixel pattern backdoor attack on non-IID FashionMNIST dataset

| Server-side Defense | Client-side Defense | MA | BA | VAR |
|---|---|---|---|---|
| None | None | 89.8 | 96.2(+0.87 / -0.79) | 0.47 |
| | LDP | 87.95 | 85.01(+0.34 / -0.63) | 0.19 |
| | WBC | 87.76 | 85.39(+1.46 / -1.23) | 1.31 |
| | LeadFL | 87.31 | 88.52(+0.76 / -0.72 ) | 0.54 |
| | LeadFL$^+$ | 86.51 | 86.47(+0.61/-0.88) | 0.55 |
| | EA-PS$^-$ | 86.47 | 86.06(+2.58 / -3.69 ) | 7 |
| | EA-PS | 86.75 | 86.46(+1.96 / -2.76 ) | 6.06 |
| CMA | None | 89.62 | 90.77(+0.58 /-0.57) | 0.33 |
| | LDP | 86.62 | 94.24(+0.53 /-0.50) | 0.19 |
| | WBC | 86.27 | 95.16(+0.61 / -0.59) | 0.35 |
| | LeadFL | 87.55 | 91.27(+1.76 / -2.53 ) | 5.03 |
| | LeadFL$^+$ | 86.77 | 87.89(+1.46/-2.23) | 6.87 |
| | EA-PS$^-$ | 86.57 | 85.52(+2.87 / -1.87 ) | 4.42 |
| | EA-PS | 86.63 | 84.96(+0.59 / -0.35 ) | 0.26 |
| CTMA | None | 89.7 | 92.92(+0.99 / -0.87) | 0.86 |
| | LDP | 87.88 | 95.91(+0.27 / -0.37) | 0.07 |
| | WBC | 88.06 | 96.4(+0.23 /-0.23) | 0.5 |
| | LeadFL | 86.95 | 87.64(+1.98 / -4.73 ) | 9.32 |
| | LeadFL$^+$ | 86.72 | 83.59(+1.29/-3.61) | 7.64 |
| | EA-PS$^-$ | 86.44 | 82.44(+1.15 / -1.68 ) | 1.45 |
| | EA-PS | 86.82 | 83.83(+1.77 / -2.42 ) | 4.72 |
| multiKrum | None | 89.35 | 57.07(+2.62 /-2.87) | 7.57 |
| | LDP | 86.29 | 32.4(+4.26 / -4.77) | 14.36 |
| | WBC | 86.35 | 43.78(+1.32/-1.72) | 1.72 |
| | LeadFL | 86.6 | 47.5(+7.29 / -8.14 ) | 66.29 |
| | LeadFL$^+$ | 86.16 | 40.96(+5.66/-4.68) | 20.36 |
| | EA-PS$^-$ | 85.64 | 34.97(+13.74 / -7.53 ) | 101.25 |
| | EA-PS | 86.03 | 35.27(+5.02 / -3.01 ) | 32.24 |
| bulyan | None | 86.75 | 56.33(+2.18 / -3.9 ) | 11.44 |
| | LDP | 85.49 | 32.25(+5.48 / -3.75) | 23.54 |
| | WBC | 85.55 | 28.02(+1.88/-3.18) | 5.1 |
| | LeadFL | 85.44 | 32.32(+8.4 / -5.7 ) | 46.74 |
| | LeadFL$^+$ | 85.41 | 33.65(+4.41/-2.15) | 26.62 |
| | EA-PS$^-$ | 85.26 | 27.29(+3.79 / -3.95 ) | 29.88 |
| | EA-PS | 84.88 | 22.13(+3.89 / -4.58 ) | 18.3 |

*Table 10.* Comparison under 9-pixel pattern backdoor attack on IID CIAFR10 dataset

| Server-side Defense | Client-side Defense | MA | BA | VAR |
|---|---|---|---|---|
| None | None | 57.4 | 79.02(+0.74/-1.35) | 1.37 |
| | LDP | 54.21 | 79.61(+1.32/-2.02) | 3.16 |
| | WBC | 53.7 | 78.5(+0.96/-0.53) | 0.7 |
| | LeadFL | 36.41 | 76.9(+7.03/-5.51) | 41.04 |
| | LeadFL$^+$ | 35.82 | 70.97(+4.22/-4.41) | 33.85 |
| | EA-PS$^-$ | 35.01 | 79.35(+4.34/-5.6) | 25.91 |
| | EA-PS | 44.17 | 77.8(+1.95/-2.93) | 6.69 |
| CMA | None | 55.87 | 58.83(+6.21/-7.96) | 52.49 |
| | LDP | 40.25 | 89.11(+1.96/-2.32) | 4.67 |
| | WBC | 38.71 | 87.15(+3.94/-5.54) | 24.39 |
| | LeadFL | 36.45 | 78.34(+3.14/-4.32) | 14.66 |
| | LeadFL$^+$ | 40.56 | 73.61(+2.58/-3.81) | 8.62 |
| | EA-PS$^-$ | 30.75 | 62.05(+4.08/-6.86) | 35.68 |
| | EA-PS | 42.9 | 63.57(+0.05/-0.04) | 0.01 |
| CTMA | None | 56.25 | 64.49(+2.26/-2.09) | 4.76 |
| | LDP | 54.29 | 90.25(+0.74/-1.43) | 1.53 |
| | WBC | 54.08 | 90.54(+2.97/-1.91) | 6.8 |
| | LeadFL | 38.2 | 64.87(+10.51/-8.57) | 93.85 |
| | LeadFL$^+$ | 38.82 | 61.75(+5.39/-7.21) | 56.73 |
| | EA-PS$^-$ | 41.42 | 57.24(+5.33/-7.38) | 43.53 |
| | EA-PS | 45.54 | 65.95(+1.55/-2.65) | 5.35 |
| multiKrum | None | 56.36 | 72.23(+9.34/-7.89) | 75.88 |
| | LDP | 53.15 | 92.02(+0.39/-0.2) | 0.11 |
| | WBC | 51.49 | 89.97(+0.88/-0.64) | 0.62 |
| | LeadFL | 42.06 | 60.63(+11.72/-13.83) | 166.51 |
| | LeadFL$^+$ | 40.29 | 63.39(+1.83/-1.72) | 5.2 |
| | EA-PS$^-$ | 32.12 | 70.03(+9.28/-10.65) | 100.78 |
| | EA-PS | 48 | 62.75(+0.43/-0.44) | 0.38 |
| bulyan | None | 55.87 | 69.29(+9.5/-5.25) | 67.94 |
| | LDP | 49.32 | 90.27(+0.7/-1.32) | 1.3 |
| | WBC | 49.7 | 89.96(+1.14/-1.02) | 1.17 |
| | LeadFL | 37.66 | 79.16(+5.23/-5.24) | 54.75 |
| | LeadFL$^+$ | 38.25 | 78.06(+4.29/-5.97) | 32.56 |
| | EA-PS$^-$ | 30.17 | 66.37(+18.15/-20.75) | 383.42 |
| | EA-PS | 30.25 | 61.3(+0.14/-0.15) | 0.04 |

*Table 11.* Comparison under 9-pixel pattern backdoor attack on non-IID CIFAR10 dataset

| Server-side Defense | Client-side Defense | MA | BA | VAR |
|---|---|---|---|---|
| None | None | 55.84 | 80.22(+1.14/-1.28) | 1.48 |
| | LDP | 52.06 | 77.4(+2.5/-2.34) | 5.89 |
| | WBC | 52.56 | 78.96(+2.5/-2.34) | 5.89 |
| | LeadFL | 45.08 | 78.78(+5.14/-6.12) | 32.38 |
| | LeadFL$^+$ | 39.11 | 78(+2.58/-1.73) | 10.52 |
| | EA-PS$^-$ | 36.19 | 78.57(+5.12/-7.18) | 40.99 |
| | EA-PS | 43.22 | 77.1(+1.46/-2.98) | 5.52 |
| CMA | None | 54.35 | 61.76(+4.34/-5.83) | 27.48 |
| | LDP | 39.22 | 85.07(+1.03/-1.25) | 1.34 |
| | WBC | 40.91 | 85.2(+2.2/-1.57) | 3.87 |
| | LeadFL | 32.79 | 61.21(+2.2/-3.83) | 11.06 |
| | LeadFL$^+$ | 36.99 | 59.51(+5.26/-4.18) | 33.69 |
| | EA-PS$^-$ | 32.76 | 63.66(+3.04/-4.93) | 18.54 |
| | EA-PS | 44.47 | 63.21(+0.13/-1.89) | 3.35 |
| CTMA | None | 55.34 | 66.7(+1.42/-2.35) | 4.17 |
| | LDP | 53.29 | 89.64(+1.72/-1.61) | 2.78 |
| | WBC | 53.75 | 92.3(+1.88/-1.81) | 3.4 |
| | LeadFL | 30.2 | 60.9(+18.7/-13.65) | 280.77 |
| | LeadFL$^+$ | 39.25 | 57.17(+4.22/-3.67) | 20.36 |
| | EA-PS$^-$ | 38.91 | 49.53(+5.44/-6.34) | 35.29 |
| | EA-PS | 44.57 | 54.82(+14.84/-9.17) | 189.97 |
| multiKrum | None | 54.97 | 68.5(+1.49/-1.01) | 1.73 |
| | LDP | 51.06 | 92.75(+0.96//-1.38) | 1.49 |
| | WBC | 50.07 | 92.13(+0.94/-0.61) | 0.69 |
| | LeadFL | 29.44 | 58.13(+5.16/-3.85) | 21.63 |
| | LeadFL$^+$ | 30.22 | 61.55(+3.95/-2.48) | 14.22 |
| | EA-PS$^-$ | 38.12 | 53.42(+2.9/-2.8) | 16.76 |
| | EA-PS | 39.81 | 52.2(+2.13/-2.44) | 8.78 |
| bulyan | None | 53.63 | 70.41(+10.4/-5.85) | 81.51 |
| | LDP | 45.64 | 90.42(+1.36/-0.75) | 1.39 |
| | WBC | 46.16 | 89.55(+1.91/-1.14) | 2.79 |
| | LeadFL | 37.3 | 64.65(+5.71/-3.98) | 25.71 |
| | LeadFL$^+$ | 39.28 | 61.76(+3.4/-4.18) | 18.55 |
| | EA-PS$^-$ | 36.35 | 71.11(+5.35/-4.2) | 23.79 |
| | EA-PS | 38.55 | 66.05(+4.42/-3.15) | 15.58 |

*Table 12.* Comparison under 1-pixel pattern backdoor attack on IID CIFAR10 dataset

| Server-side Defense | Client-side Defense | MA | BA | VAR |
|---|---|---|---|---|
| None | None | 55.73 | 39.25(+5.91/-4.37) | 28.22 |
| | LDP | 53.64 | 49.19(+2.06/-3.57) | 9.64 |
| | WBC | 52.77 | 44.23(+8.8//-5.3) | 58.88 |
| | LeadFL | 40.67 | 53.09(+11.21/-13.11) | 158.59 |
| | LeadFL$^+$ | 39.36 | 54.79(+12.71/-10.88) | 141.63 |
| | EA-PS$^-$ | 38.38 | 61.14(+1.92/-2.42) | 4.9 |
| | EA-PS | 39.79 | 54.52(+3.59/-6.83) | 34.95 |
| CMA | None | 54.05 | 21.14(+2.81/-2.56) | 7.26 |
| | LDP | 37.23 | 33.4(+2.21/-3.01) | 7.3 |
| | WBC | 34.32 | 40.3(+5.45/-7.55) | 45.6 |
| | LeadFL | 40.14 | 46.54(+17.35/-17.36) | 301.31 |
| | LeadFL$^+$ | 41.13 | 54.79(+12.71/-10.88) | 141.63 |
| | EA-PS$^-$ | 34.64 | 44.71(+23.3/-16.13) | 427.22 |
| | EA-PS | 39.8 | 32.58(+0.09/-0.1) | 0.02 |
| CTMA | None | 56.54 | 30.95(+0.55/-0.5) | 0.28 |
| | LDP | 54.24 | 72.49(+1.98/-1.69) | 3.44 |
| | WBC | 53.98 | 73.17(+2.25/-1.45) | 3.92 |
| | LeadFL | 49.31 | 64.46(+9/-9.01) | 162.16 |
| | LeadFL$^+$ | 42.56 | 54.58(+7.25/-6.16) | 48.1 |
| | EA-PS$^-$ | 34.07 | 50.36(+28.63/-14.5) | 614.8 |
| | EA-PS | 36.24 | 42.62(+0.49/-0.34) | 0.19 |
| multiKrum | None | 55.71 | 58.08(+2/-3.31) | 8.33 |
| | LDP | 51.81 | 73.06(+3.38/-1.8) | 8.56 |
| | WBC | 52.05 | 76.63(+2.64/-2.08) | 5.8 |
| | LeadFL | 30 | 64.26(+15.5/-29.21) | 640.64 |
| | LeadFL$^+$ | 32.88 | 61.72(+7.98/-5.76) | 62.41 |
| | EA-PS$^-$ | 31.05 | 67.4(+11.3/-11.23) | 126.97 |
| | EA-PS | 40.3 | 52.5(+2.4/-2.7) | 15 |
| bulyan | None | 54.06 | 60.57(+0.77/- 1.02) | 0.85 |
| | LDP | 49.16 | 76.12( +0.89/- 1.42) | 1.53 |
| | WBC | 48.56 | 76.48(+0.71/-0.72) | 1.02 |
| | LeadFL | 34.75 | 63(+16.85/-13.28) | 236.4 |
| | LeadFL$^+$ | 33.76 | 66.14(+8.77/-7.39) | 120.55 |
| | EA-PS$^-$ | 29.37 | 59.15(+21.54/-23.12) | 500.37 |
| | EA-PS | 33.88 | 63.74(+9.04/-10.51) | 97.22 |

*Table 13.* Comparison under 1-pixel pattern backdoor attack on non-IID CIFAR10 dataset

| Server-side Defense | Client-side Defense | MA | BA | VAR |
|---|---|---|---|---|
| None | None | 56.17 | 41.3(+8.58/-7.35) | 64.56 |
| | LDP | 52.29 | 49.51(+1.67/-1.67) | 2.78 |
| | WBC | 54.04 | 45.37(+4.17/-5.91) | 27.72 |
| | LeadFL | 37.64 | 51.84(+7.01/-8.62) | 63.03 |
| | LeadFL$^+$ | 38.81 | 58.33(+3.21/-3.7) | 10.55 |
| | EA-PS$^-$ | 37.8 | 54.74(+3.86/-5.12) | 21.37 |
| | EA-PS | 40.31 | 50.6(+3.84/-4.05) | 15.6 |
| CMA | None | 54.79 | 21.31(+0.74/-1.34) | 1.34 |
| | LDP | 37.52 | 37.18(+3.07/-4.94) | 18.7 |
| | WBC | 30.25 | 35.95(+19.66/-13.88) | 306.25 |
| | LeadFL | 37.39 | 31.47(+1.3/- 1.64) | 2.35 |
| | LeadFL$^+$ | 36.2 | 36.46(+3.29/-2.8) | 30.19 |
| | EA-PS$^-$ | 35.98 | 37.66(+9.34/-16.43) | 83.36 |
| | EA-PS | 41.36 | 33.18(+7.08/-10.69) | 91.01 |
| CTMA | None | 55.69 | 29.58(+3.73/-4.44) | 17.04 |
| | LDP | 52.81 | 75.64(+1.37/-2.16) | 3.6 |
| | WBC | 52.75 | 75.24(+1.04/-1.24) | 1.33 |
| | LeadFL | 30.34 | 43.3(+16.8/-9.77) | 213.57 |
| | LeadFL$^+$ | 40.5 | 42.61(+4.71/-3.92) | 40.6 |
| | EA-PS$^-$ | 38.96 | 39.6(+8.29/-9.54) | 57.13 |
| | EA-PS | 43.96 | 40.01(+3.46/-2.09) | 9.12 |
| multiKrum | None | 53.81 | 47.25(+6.4/-3.99) | 31.3 |
| | LDP | 49.94 | 75.55(+3.72/-3.44) | 12.86 |
| | WBC | 49.54 | 76.56(+2.66/-4.84) | 17.64 |
| | LeadFL | 30.23 | 61.74(+3.92/-2.33) | 11.71 |
| | LeadFL$^+$ | 33.4 | 56.84(+3.89/-3.72) | 26.28 |
| | EA-PS$^-$ | 36.63 | 45.2(+2.41/-1.61) | 4.54 |
| | EA-PS | 31.33 | 52.5(+2.49/-4.46) | 15 |
| bulyan | None | 55.54 | 56.28(+4.93/-6.65) | 35.72 |
| | LDP | 44.54 | 79.19(+0.86/-0.78) | 0.68 |
| | WBC | 44.45 | 78.18(+2.79/-2.94) | 8.22 |
| | LeadFL | 32.31 | 65.96(+9.02/-7.79) | 71.8 |
| | LeadFL$^+$ | 39.92 | 56.02(+7.29/-8.06) | 92.72 |
| | EA-PS$^-$ | 40.5 | 58.19(+1.59/-1.65) | 2.63 |
| | EA-PS | 43.12 | 59.11(+1.78/-3.18) | 7.61 |

*Table 14.* Comparison of different $\beta$ on FashionMNIST with IID/non-IID settings under 1/9-pixel backdoor attack.

| (mean(%)\var($10^{-4}$)) | | IID | | | | | non-IID | | | | |
|---|---|---|---|---|---|---|---|---|---|---|---|
| | $\beta$ | 0.01 | 0.03 | 0.05 | 0.07 | 0.09 | 0.01 | 0.03 | 0.05 | 0.07 | 0.09 |
| 9-pixel | none | 90.24/0.05 | **87.08/6.17** | 95.87/0.61 | **88.7/4.36** | 89.38/10.78 | 87.23/0.44 | **86.47/3.69** | **87.79/0.21** | 87.97/7.68 | 86.96/2.31 |
| | Multikrum | 32.45/51.18 | 25.81/39.85 | **22.38/84.09** | **18.29/0.97** | 24.13/30.63 | 16.17/251.97 | **6.61/8.09** | **7.39/13.4** | 17.55/1.42 | 24.13/30.63 |
| | Bulyan | 21.4/29.19 | 29.25/4.77 | 33.15/175.49 | **19.58/0.01** | **16.24/59.42** | 11.39/10.38 | 6.85/26.68 | **5.19/4.25** | 13.01/48.52 | **6.3/4.24** |
| 1-pixel | none | 91.96/0.12 | 91.90/0.92 | **89.09/0.07** | 87.28/0.44 | 91.60/3.1 | 89.36/19.39 | **86.46/6.06** | 89.35/0.98 | 89.81/0.94 | **87.21/1.85** |
| | Multikrum | **45.58/0.66** | 59.74/176.35 | **44.65/6.67** | 57.04/118.49 | 46.07/74.95 | **39.50/0.64** | **31.59/98.35** | **36.42/132.46** | 51.47/64.03 | 40.69/41.37 |
| | Bulyan | **45.68/189.3** | 54.14/32.44 | **39.62/14.76** | 47.81/184.62 | 55.9/116.15 | 33.06/32.33 | **26.52/0.06** | 30.72/56.31 | **25.40/221.80** | 27.95/1.29 |

*Table 15.* Comparison of different $\alpha$ on FashionMNIST with IID/non-IID settings under 1/9-pixel backdoor attack.

| $\alpha$(mean(%)/var($10^{-4}$)) | server-defense | 0.1 | 0.2 | 0.3 | 0.4 | 0.5 | 0.6 | 0.7 | 0.8 | 0.9 |
|---|---|---|---|---|---|---|---|---|---|---|
| IID — 9-pixel | MultiKrum | 8.85/1.93 | 23/3.96 | 35.74/1.77 | 28.46/10.25 | 13.05/1.32 | 27.73/12.65 | 30.61/1.55 | 25.86/10.09 | 19.75/5.06 |
| | Bulyan | 26.87/76.91 | 7.96/1.48 | 32.39/59.97 | 42.03/139.27 | 36.81/68.19 | 41.97/136.82 | 33.44/0.33 | 47.14/257.44 | 67.64/37.09 |
| IID — 1-pixel | MultiKrum | 69.27/15.19 | 43.07/52.73 | 42.03/50.85 | 33.58/5.43 | 37.05/2.83 | 39.29/7.25 | 48.45/4.44 | 38.64/3.88 | 48.12/3.25 |
| | Bulyan | 31.26/183.19 | 54.07/125.02 | 36.05/145.32 | 34.16/97.46 | 41.12/65.53 | 51.71/95.86 | 73.17/143.11 | 80.37/25.9 | 85.27/0.12 |
| non-IID — 9-pixel | MultiKrum | 5.34/15.48 | 23.33/8.24 | 8.82/3.22 | 10.34/7.22 | 10.49/20.36 | 6.83/14.09 | 10.34/9.32 | 13.34/4.65 | 16.89/12.81 |
| | Bulyan | 20.14/14.45 | 20.04/21.26 | 7.32/1.14 | 25.5/1.16 | 20.33/6.58 | 19.27/41.37 | 25.57/5.44 | 28.53/4.66 | 30.45/2.12 |
| non-IID — 1-pixel | MultiKrum | 22.39/19.34 | 25.39/16.47 | 39.45/10.26 | 36.84/1.55 | 40.07/7.45 | 36.57/2.15 | 42.38/0.01 | 26.67/9.8 | 42.07/13.23 |
| | Bulyan | 34.66/30.25 | 42.63/275.9 | 18.99/89.01 | 32.57/5.43 | 30.04/31.08 | 30.51/29.67 | 36.45/18.98 | 41.44/28.32 | 42.63/35.36 |

*Table 16.* Comparison of different linear ratio in $\lambda$ on FashionMNIST with IID/non-IID settings under 1/9-pixel backdoor attack.

| (mean(%)/var($10^{-4}$)) | | IID | | | | | non-IID | | | | |
|---|---|---|---|---|---|---|---|---|---|---|---|
| linear ratio in $\lambda$ | | 0.1 | 0.3 | 0.5 | 0.7 | 0.9 | 0.1 | 0.3 | 0.5 | 0.7 | 0.9 |
| 9-pixel | MultiKrum | 45.02/8.3 | 32.75/8.33 | 10.83/18.66 | 22.86/12.18 | 36.58/16.27 | 16.8/6.78 | 7.81/0.55 | 8.32/6.44 | 10.54/4.81 | 11.73/11.93 |
| | Bulyan | 21.14/13.68 | 15.94/1.08 | 17.67/1.55 | 19.47/10.61 | 27.96/30.29 | 4.81/0.04 | 20.02/14.3 | 7.59/4.32 | 24.47/22.88 | 10.29/7.51 |
| 1-pixel | MultiKrum | 49.96/5.76 | 61.99/7.16 | 47.25/2.54 | 40.34/1.38 | 49.06/1.01 | 54.43/3.83 | 36.39/5.05 | 36.86/1.57 | 39.96/4.42 | 27.84/3.41 |
| | Bulyan | 30.64/1.47 | 46.92/13.68 | 55.78/5.51 | 31/0.46 | 42.94/3.37 | 46.85/1.72 | 30.7/2.74 | 29.88/9.24 | 41.56/4.38 | 28.34/2.4 |

*Table 17.* Comparison of different $\gamma$ on FashionMNIST with IID/non-IID settings under 1/9-pixel backdoor attack.

| mean%/var$10^{-4}$ | | IID | | | | | non-IID | | | | |
|---|---|---|---|---|---|---|---|---|---|---|---|
| $\gamma$ | | 0.01 | 0.03 | 0.05 | 0.07 | 0.09 | 0.01 | 0.03 | 0.05 | 0.07 | 0.09 |
| 9-pixel | MultiKrum | 22.19/17.57 | 42.16/4.42 | 34.44/9.69 | 22.94/1.98 | 27.09/9.35 | 13.49/4.82 | 7.39/2.43 | 28.7/7.3 | 4.39/0.64 | 15.59/8.01 |
| | Bulyan | 34.01/67.96 | 4.55/0.04 | 19.17/31.21 | 10.06/6.71 | 37.61/4.43 | 7.21/0.8 | 7.8/3.24 | 4.47/1.22 | 4.25/0.65 | 9.57/1.02 |
| 1-pixel | MultiKrum | 50.89/0.79 | 40.24/2.11 | 48.95/0.78 | 46.17/7.74 | 48.96/7.2 | 34.32/5.37 | 24.37/0.98 | 27.99/7.66 | 56.77/6.3 | 20.44/7.25 |
| | Bulyan | 43.74/2.29 | 36.78/0.83 | 44.72/17.42 | 40.9/7.87 | 23.76/2 | 29.57/4.60 | 17.61/2.1 | 25.85/9.09 | 25.06/7.05 | 44.9/3.68 |