# OpenReview forum: "EA-PS: Estimated Attack Effectiveness based Poisoning Defense in Federated Learning under Parameter Constraint Strategy"
_ICML.cc/2025/Conference — Submitted to ICML 2025_

### Official Review · Reviewer_Djsj · 2025-03-12

**Overall Recommendation:** 3

**Summary:**

This paper proposes a client-side defense methond in federated learning, EA-PS, that constrains the pertubation range of local parameters while minimizing the impact of attacks by forming the problem into an optimization problem. This paper further provides convergence and robustness analysis. This paper validates its algorithm through experiments.

**Claims And Evidence:**

This paper claims that with EA-PS, combined with server-side defense method, can achieve robust and stable performance under attack. The claims are clearly supported through theoretical results and empirical results.

**Essential References Not Discussed:**

There are no essential references not discussed to my knowledge.

**Experimental Designs Or Analyses:**

I have checked the experiments compared proposed methods against other baselines, varying \beta, varying \alpha, different \lambda and different \gamma. No issues discovered.

**Methods And Evaluation Criteria:**

The experiments are extensive with varying parameters of the algorithms, supporting the claim of the paper.

**Other Comments Or Suggestions:**

None.

**Other Strengths And Weaknesses:**

Strengths: Experiments are extensive with strong theoretical guarantees.
Weaknesses: Preliminary knowledge isn't explanied enough, make it hard to follow the paper. For example, why A_t - A_{t-1} can be interpreted as long-lasting attacks? what is the definition of long-lasting attacks?
Figure 2 didn't illustrate the idea of parameter constraint strategy.

**Questions For Authors:**

1. why can we assume \lambda is a linear set of A? What will be sacrifice with this assumption?
2. How is \tilde{H} calculated?

**Relation To Broader Scientific Literature:**

The key contributions of the paper is to constraint the local parameter updating in federated learning under attack, which reduce the variance during learning process.

**Theoretical Claims:**

I have checked the proof of Theorem 4.1, no issues discovered.

---

> ### Author Rebuttal · Authors · 2025-03-30
>
> Thank you for your thorough analysis and constructive feedback on our paper. We appreciate the opportunity to clarify the points raised and to provide additional insights into our research.
>
> >  1. What is the definition of long-lasting attacks? Why $A_t - A_{t-1}$ can be interpreted as long-lasting attacks?
>
> Response:
>
> - We appreciate the opportunity to clarify the definition of long-lasting attacks. In our work, we follow the FL-WBC's observations on the long-lasting attack effect. The definition of "long-lasting" in the long-lasting attack is to describe the effects of an attack in the current round that can persist through multiple rounds of training. Therefore, there exists a slight misleading in our work. The correction is as follows: “long-lasting attack” to "attack with long-lasting effects ".
>
> -  $A_t-A_ {t-1}$ in our work is "the difference of different rounds in the coefficient of attack effects". $A_t$ is defined as "the coefficient of attack impact between two rounds". $A_t-A_ {t-1}$ can form a chain structure to better measure the accumulation of attack impact in different periods (i.e., long-lasting attack effects). For example, in our work, Theorem 4.1 shows that minimizing $A_t-A_ {t-1}$ yields a smaller optimization upper bound than a traditional method such as LeadFL; Theorem 5.2: Certified Radius analysis shows that the reduction of $A_t-A_ {t-1}$ improves the robustness of the model against long-term attack effects. The experimental results also support the above proof.
>
> > 2. Figure 2 didn't illustrate the idea of parameter constraint strategy.
>
> Response:
>
> - We believe that Figure 2 illustrates the idea of the parameter constraint strategy. Based on the last paragraph of section 3, we will provide additional insights into our strategy. The goal of the parameter constraint strategy is to enhance the stability of poisoning attack defense by constraining the perturbation range of model parameters. The key components are 1)Optimized Manifold Space $A$; 2) Unit Space $I$; 3) Rank Constraint $λ$.
>
> - Optimized Manifold Space $A$ represents the unconstrained parameter space of the model, which may involve high-dimensional or complex parameter distributions. In this space, malicious attacks (e.g., backdoor attacks) can create long-lasting effects through parameter perturbations. As illustrated in equations (formalized as equation (16): $I=B^{-1}AB$), the manifold space $A$ is mapped into a simpler, low-dimensional unit space (Unit Space $I$).  By constraining parameter perturbations within a bounded region (Rank Constraint $λ$), the strategy suppresses the cumulative effects of adaptive or persistent attacks (formalized as equation (17): $AB=λB$). This ensures stable defense performance under long-lasting attack effects.
>
> > 3. Why can we assume $\lambda$ is a linear set of $A$? What will be sacrifice with this assumption?
>
> Response:  For the reason why we can assume $\lambda$ is a linear set of $A$, based on the references and descriptions on page 5 of our work, we will provide additional insights into it. Firstly, we assume that $λ$ is a linear set of $A$ mainly because the linear decision rule can transform complex uncertainty descriptions into a more tractable linear form, thus yielding a computationally solvable robust optimization model. Specifically, the linear assumption simplifies the complex parameter constraints into a linear combination of historical information. Moreover, linear approximations facilitate the simplification of proofs for convergence and robustness guarantees. However, setting $λ$ as a linear combination of $A$ means ignoring possible nonlinear relationships, which may result in suboptimality according to the linear decision rule (Bertsimas et al., 2019), which has noted on page 5 of our work.
>
> > 4. How is $H$ calculated?
>
> Response: We appreciate the opportunity to add details about how $H$ is calculated. We will change the equation of $H_{t,e}^k$ in our work (section 4.2) to illustrate the details of how $H$ is calculated, as follows.
>
> $H_{t,e}^k \overset{\bigtriangleup }{\underset{}{=}}\bigtriangledown ^2 F(
> \theta _{t,e}^k)= (θ _{t,e+1}^k-θ _{t,e}^k-∆θ _{t,e}^k)/ η_t.$
>
> We hope this response adequately addresses your points and welcome the fruitful discussion. We are thankful for the contribution to the manuscript's refinement.

---

### Official Review · Reviewer_qmsf · 2025-03-13

**Overall Recommendation:** 3

**Summary:**

To combat persistent adaptive attacks, the authors propose EA-PS, a client-side defense that enhances server-side methods for robust, stable performance. By limiting attack impact and constraining local parameter perturbations, EA-PS mitigates backdoor poisoning. Theoretically, it achieves a lower upper bound, smaller certified radius, and larger convergence upper bound. Evaluations on FashionMNIST and CIFAR-10 show EA-PS reduces attack success rates by up to 14.9% and improves stability with up to 40% lower variance compared to other client-side defenses.

**Claims And Evidence:**

The paper presents strong empirical and theoretical evidence supporting the effectiveness of the proposed method. However, two key aspects are missing: (1) The cost of implementing client-side defense, such as communication overhead, should be compared to pure server-side defense. (2) The efficiency of the proposed method is not thoroughly evaluated—while convergence results provide some insight, empirical experiments are needed. Specifically, how much additional time does the client-side defense require compared to standard FedAvg? Additionally, it would be beneficial to show model accuracy and backdoor accuracy throughout the FL process to illustrate whether this defense slows down main task training, which is just as crucial as security in practice. Overall, most of my concerns are from empirical perspective, I appreciate the authors offer theoretical guarantees.

**Essential References Not Discussed:**

N/A

**Experimental Designs Or Analyses:**

See "Methods And Evaluation Criteria"

**Methods And Evaluation Criteria:**

1. I will suggest adding different type of backdoor attacks (e.g., distributed trigger, adaptive backdoor) as baselines, except for only using fixed pattern since the theoretically results suggest a general defense.
2. Multikrum and Bulyan are designed to defend against model poisoning attacks. Defenses that include a post-training stage, such as CRFL, should also be considered as baselines.

**Other Comments Or Suggestions:**

It is better to clarify the defender's knowledge and ability considering the difference between client-side defense and server-side defense. To make it more practical, how clients and/or sever exchange knowledge, information should be specified.

**Other Strengths And Weaknesses:**

Minor: Backdoor attacks are typically considered a specific type of targeted attack in previous FL security papers, which is slightly inconsistent with Section 2.1.

**Questions For Authors:**

The attack impact measures the differences between two rounds. I wonder why multiple rounds are not considered, as in a real-world FL system, malicious clients may not be selected every round. Additionally, I suggest measuring the tradeoff between security and training speed, as the current optimization goal may slow down the training process.

**Relation To Broader Scientific Literature:**

N/A

**Theoretical Claims:**

N/A

---

> ### Author Rebuttal · Authors · 2025-03-30
>
> Thank you for your thorough review and valuable feedback on our work.
>
> >  1. The communication overhead should be compared to pure server-side defense.
>
> Response: We'd like to address the concern regarding the communication overhead in our work. Since nothing but the parameter constraint strategy is used in this work, the communication overhead is the same as pure server-side defense per round.
>
> > 2. I will suggest adding distributed trigger, adaptive backdoor, and CRFL as baselines.
>
> Response:  For distributed trigger and adaptive backdoor, we added DBA[2] and A3FL[1].  We also added CRFL to compare with Multi-Krum and Bulyan.  The results are as follows.
>
> |  (%)  |  |  CIFAR-10 |  |   |   |   |
> |--|--|--|--:|:--|--:|:--|
> |         |         |             | DBA   |       | A3FL  |       |
> |         | Client  | Server | MA| BA| MA| BA| MA | BA |
> | IID    | EA-PS   | MultiKrum |34.51|76.72|32.15|57.96|
> |         |         | Bulyan |33.72|80.26|32.87|59.37|
> |         |         | CRFL   |27.49|12.53|27.29|26.42|
> |         | Lead-FL | MultiKrum  |35.68|77.08|33.96|58.49|
> |         |         | Bulyan|34.94|49.14|32.65|52.8|
> |         |         | CRFL  |26.89|24.73|27.18|39.83|
> | Non-IID| EA-PS   | MultiKrum   |34.18|38.94|32.64|47.49|
> |         |         | Bulyan |35.24|30.02|34.74|41.38|
> |         |         | CRFL  |26.29|10.59|25.82|13.67|
> |         | Lead-FL | MultiKrum  |35.18|45.95|34.53|53.62|
> |         |         | Bulyan |35.01|41.64|33.59|49.86|
> |         |         | CRFL  |26.41|11.89|26.25|17.84|
>
>
>
> > 3. How much additional time does the client-side defense require compared to standard FedAvg?
>
>  Response: As convergence analysis proved, our work is slightly inefficient compared with other methods.   We appreciate your suggestion to add time overheads (seconds average round), and the experimental results on CIFAR-10 with IID distribution and FEMNIST with natural non-IID distribution are as follows.
>
> | Time(s/r) |CIFAR-10|  |    |   |FEMNIST  |  |    |    |
> |--|--:|:--|:--|:--|--:|--:|--:|--:|
> |         | FedAvg| Krum | Bulyan | CRFL | FedAvg  | Krum | Bulyan | CRFL |
> | EA-PS   |27.98|28.66|28.21|45.13|80.75|82.63|78.89|118.51|
> | Lead-FL |22.55|23.11|22.83|40.62|56.51|58.74|58.02|109.36|
> | FL-WBC |20.95|21.34|21.27|38.32|44.17|50.44|49.38|98.27|
> | NULL |19.74|20.94|20.61|37.68|43.17|47.55|46.57|97.46|
>
> > 4.  Backdoor attacks is slightly inconsistent with Section 2.1.
>
> Response:   We appreciate the opportunity to change the  "slightly inconsistent" expression that "**one of the specific types of** targeted attacks (known as backdoor attacks )" in section 2.1.
>
> >   5. It would be beneficial to show model accuracy and backdoor accuracy to illustrate whether this defense slows down main task training.
>
> Response: Details of MA (Main-task Accuracy) are in the Appendix.  We will extract the MA to the experiment section.
>
> >  6. The attack impact measures the differences between two rounds. I wonder why multiple rounds are not considered, as in a real-world FL system, malicious clients may not be selected every round.
>
> Response:  We appreciate the opportunity to clarify the "multiple rounds attack impact". Our work follows previous works setting as in Lead-FL and FL-WBC, where "In each adversarial round, malicious clients are randomly selected and participate in the training". We will highlight this in the experimental setting.
>
> > 7.  It is better to clarify the defender's knowledge and ability considering the difference between client-side defense and server-side defense. To make it more practical, how clients and/or sever exchange knowledge, information should be specified.
>
> Response:  1) Only aggregation information is exchanged between clients and the server. 2)  To clarify  "the defender's knowledge and ability, considering the difference between client-side defense and server-side defense", we will add the comparisons in a table as follows.
> | Component   | Client-Side Defense (Ours)  | Server-Side Defense  |
> |--|--|--|
> | Knowledge   | Local model parameters and gradients；Local training data distribution  |  Global aggregated model；Aggregated update statistics (e.g., gradient norms)  |
> | Capability  |  Can apply local parameter masking/smoothing; Cannot modify server aggregation logic | Can modify aggregation rules (e.g., clip gradients, weight averaging)       |
> | Assumptions |Clients may be malicious ；Server is honest  | Server is fully trusted；Clients may be malicious   |
>
> The added code will still open-source to the original link in the manuscript. We are grateful for the chance to discuss our work's improvement, and wish to thank you again for your valuable input.
>
> Reference:
>
> [1]Zhang, Hangfan et al. “A3FL: Adversarially Adaptive Backdoor Attacks to Federated Learning.” Neural Information Processing Systems (2023).
>
> [2] DBA: Xie, et al. "Distributed Backdoor Attacks against Federated Learning." International Conference on Learning Representations (2020)

---

### Official Review · Reviewer_dbKW · 2025-03-13

**Overall Recommendation:** 3

**Summary:**

This paper proposes EA-PS (Estimated Attack Effectiveness-based Poisoning Defense with Parameter Constraint Strategy), a client-side defense designed to constrain the perturbation range of local parameters while minimizing the impact of attacks.
The authors prove that our methods have an efficiency guarantee with a lower upper bound, a robustness guarantee with a smaller certified radius, and a larger convergence upper bound.

**Claims And Evidence:**

- efficiency guarantee with a lower upper bound
Evidence:
4.2 and Appendix A.2

- a robustness guarantee with a smaller certified radius
5.3 and Appendix A.4 Theoretical analysis

- a larger convergence upper bound
5.2 and Appendix A.3 Theoretical analysis

**Essential References Not Discussed:**

I don't know well the related literature for client-side defense.
But I think the attacks evaluated are limited, as mentioned above

**Experimental Designs Or Analyses:**

As mentioned before:
Dataset: FashionMNIST and CIFAR10 datasets under both IID and non-IID settings. I think the datasets, although widely used in FL, do not contain real-world noniid and is too simple. can the author also consider dataset like FEMINIST, which contains nature non-iid?
The attack method used here is only one pattern. I encourage the author to check the performance also for untargeted attack, and other targeted attack, such as follows:

[1] ] Xiaoyu Cao and Neil Zhenqiang Gong. 2022. Mpaf: Model poisoning attacks to federated learning based on fake clients.
[2] DBA: Distributed Backdoor Attacks against Federated Learning

**Methods And Evaluation Criteria:**

1. introduce an enhanced objective function (EA-PS−)
2. propose a client-based defense approach named Estimated Attack Effectiveness based Poisoning Defense method under Parameter Constraint Strategy (EA-PS). It minimizes the long-lasting backdoor attack effect with a parameter constraint strategy to enhance stability
by constraining the perturb range in the parameter space

Evaluation: main task accuracy(MA), backdoor accuracy(BA)  I think the evaluation metric is reasonable
Dataset: FashionMNIST and CIFAR10 datasets under both IID and non-IID settings. I think the datasets, although widely used in FL, do not contain real-world noniid and is too simple. can the author also consider dataset like FEMINIST, which contains nature non-iid?
The attack method used here is only one pattern. I encourage the author to check the performance also for untargeted attack.

**Other Comments Or Suggestions:**

See above.

**Other Strengths And Weaknesses:**

Table 1 caption is not quite clear benign accuracy / （attack success rate）？

**Questions For Authors:**

See above

**Relation To Broader Scientific Literature:**

It supplements the client-side poisoning defense with theoretical guarantees.

**Theoretical Claims:**

- efficiency guarantee with a lower upper bound
Evidence:
4.2 and Appendix A.2

- a robustness guarantee with a smaller certified radius
5.3 and Appendix A.4 Theoretical analysis

- a larger convergence upper bound
5.2 and Appendix A.3 Theoretical analysis

I don't see flaws in the theoretical analysis , yet I'm not an expert of theory, please refer to other reviewer's suggestions.

---

> ### Author Rebuttal · Authors · 2025-03-30
>
> Thank you for your recognition of our work and for your insightful comments.
> > 1. Table 1 caption is not quite clear benign accuracy / （attack success rate）？
>
> Response:  The metric used in Table 1 is backdoor accuracy, which is the attack success rate for backdoor attacks.  We will change it to  "backdoor accuracy" in the revision.
>
> >2. Can the author consider FEMINIST, which contains nature non-iid? I encourage the author to check the performance for untargeted attack (MPaf) and other targeted attack (DBA).
>
> Response:
> - We appreciate your suggestion to include the nature non-iid dataset (FEMINIST), MPaf, and DBA. In addition to the methods suggested above, we also added  Spectrum (targeted) and  Label-Flip (untargeted) to further enhance the experiment.  The results are as follows.
>
>
>
> | (%)  |        |             |       |            |       |  FEMINIST  |        |       |          |         |            |
> |--|--|--:|:--|--:|:--|--:|:--|--:|:--|--|--|
> |   |        | 1-pixel |  | 9-pixel |       | Spectrum|(our)       | DBA   | (suggested)   | Label-Flip (our) |   MpAf(suggested) |
> | client  | server | MA    | BA    | MA      | BA    | MA       | BA    | MA    | BA    | MA    | MA    |
> | EA-PS  |MultiKrum|87.76|43.2|88.48|53.84|87.92|4.24|87.57|18.79|87.95|88.01|
> |       | Bulyan | 86.25| 49.43|84.76|66.43|86.84|4.082|87.67|7.03|87.81|87.47|
> | Lead-FL | MultiKrum|88.31|	65.75|88.38|59.01|88.57|4.61|88.32|25.24|87.7|87.94|
> |       | Bulyan  |88.25|65.09|87.81|76.62|88.53|4.32|87.17|9.35|87.44|87.17|
>
> - We also apply the suggested attack methods and our added methods on the CIFAR-10 dataset to further illustrate the performance. The results are as follows.
>
> |  (%)     |     |  CIFAR-10   |  |   |     |      |     |    |
> |--|--|--|--:|:--|--:|:--|--|--|
> |         |         |        |  Spectrum  | (our)   | DBA | (suggested)     | Label-Flip (our) |MpAf (suggested)|
> |         | Client  | Server | MA       | BA    | MA    | BA     | MA   | MA     |
> | IID    | EA-PS   | MultiKrum  |32.41|73.06|34.51|76.72|13.39|14.15|
> |         |         | Bulyan |32.18|49.76|33.72|80.26|14.8|14.94|
> |         | Lead-FL | MultiKrum  |33.95|76.77|35.68|77.08|10.9|10.47|
> |         |         | Bulyan |33.44|40.85|34.94|49.14|14.42|14.27|
> | Non-IID| EA-PS   | MultiKrum   |33.75|46.12|34.18|38.94|15.9|15.67|
> |         |         | Bulyan |34.81|40.13|35.24|30.02|16.08|15.94|
> |         | Lead-FL | MultiKrum   |33.95|55.42|35.18|45.95|13.42|13.19|
> |         |         | Bulyan |32.1|41.33|35.01|41.64|16.24|15.35|
>
> -  It's important to note that none of the existing client-side defense methods focus on untargeted attacks. Through added experiments, we found that although our method struggles to defend against untargeted attacks, it still slightly outperforms the state - of - the - art client - side defense methods.
>
> - For new target attacks, our method outperforms the state - of - the - art client - side defense methods with server-side defense methods.
>
> The added code will still open-source to the original link in the manuscript.
>
> We hope this response has addressed your concerns effectively. We are grateful for your valuable input.
>
> References
>
> [1] Wang, Tong et al. “An Invisible Black-Box Backdoor Attack Through Frequency Domain.” European Conference on Computer Vision (2022). (Spectrum)
>
> [2]Zhang, Mengmei et al. “Adversarial Label-Flipping Attack and Defense for Graph Neural Networks.” 2020 IEEE International Conference on Data Mining (ICDM) (2020): 791-800. (Label-Flip)

---

### Decision · Program_Chairs · 2025-05-01

**Decision:**

Reject

**Comment:**

This paper proposes EA-PS (Estimated Attack Effectiveness-based Poisoning Defense with Parameter Constraint Strategy), a client-side defense designed to constrain the perturbation range of local parameters to minimizing the impact of targeted attacks. The paper provide both theoretical and empirical evidence for the effectiveness of the proposed defense.

Feedback for the authors: please add the new experiments to the paper, as well as discussions regarding computational cost, (in)-effectiveness against untargeted attacks.

With the above changes, this paper can make a valuable contribution to the literature of provably efficient defenses against poisoning attack in federated learning.